

# Airborne Lidar Measurements of Aerosol and Ozone Above the Canadian Oil Sands Region

Monika Aggarwal[1], James Whiteway[1], Jeffrey Seabrook[1], Lawrence Gray[1], Kevin Strawbridge[2], Peter Liu[2], Jason O'Brien[2], Shao-Meng Li[2], Robert McLaren[3]

[1]York University, Centre for Research in Earth and Space Science, Toronto, M3J 1P3, Canada
[2]Evironment and Climate Change Canada, Air Quality Processes Research Section, Toronto, M3H 5T4, Canada
[3]York University, Centre for Atmospheric Chemistry, Toronto, M3J 1P3, Canada

*Correspondence to*: James Whiteway (whiteway@yorku.ca)

**Abstract.** Aircraft based lidar measurements of atmospheric aerosol and ozone were conducted to study air pollution from the oil sands extraction industry in northern Alberta. Significant amounts of aerosol were observed in the polluted air within the surface boundary layer, up to heights of 1 km to 1.5 km above ground. The ozone mixing ratio measured in the polluted boundary layer air was equal to or less than the background ozone mixing ratio, in the range of 10 ppbv to 35 ppbv. On one of the flights, the
lidar measurements detected a layer of forest fire smoke above the surface boundary layer in which the ozone mixing ratio had a maximum value of 70 ppbv. Measurements of the linear depolarization ratio in the aerosol backscatter were obtained with a ground based lidar and this aided in the discrimination between the separate emission sources from industry and forest fires. The retrieval of ozone abundance from the lidar measurements required the development of a method to account for the interference from
the substantial aerosol content within the surface boundary layer.

## 1 Introduction

The oil sands mining and upgrading facilities in northern Alberta are a known source of air pollution (Simpson et al., 2010; Liggio et al., 2016). The Joint Canada-Alberta Implementation Plan for Oil Sands Monitoring (JOSM) was organized by Environment and Climate Change Canada (ECCC) to monitor
the impact of oil sands emissions on the quality of air, water, land, and wildlife (Abbatt et al., 2011). This paper concerns the methodology and results of airborne lidar measurements of aerosol and ozone that were contributed to the JOSM field campaign during August 2013. A lidar instrument from York

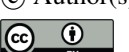



University was installed on a Twin Otter aircraft in order to complement the main research aircraft for the JOSM flight campaign, the Convair-580 from the National Research Council of Canada (Gordon et al., 2015; Liggio et al., 2016).

Emissions from the oil sands facility stacks and mining operations include nitrogen oxides (NO and $NO_2$), sulfur dioxide ($SO_2$), sulfate ($SO_4$), methane ($CH_4$), carbon dioxide ($CO_2$), carbon monoxide (CO), particulate matter, volatile organic compounds (VOC), and secondary organic aerosol (SOA) (Simpson et al., 2010; Davies, 2012; Howell et al., 2014; Liggio et al., 2016; Li et al., 2017; Baray et al., 2017). Large amounts of $NO_x$ and $SO_2$ emissions within the oil sands region originate from the

upgrading processes and high temperature combustion of oil, gasoline, and coal. The facilities are surrounded by boreal forests which are a natural source of biogenic VOC emissions and also smoke emissions from forest fires. Enhancements in ozone are well known to occur in urban air pollution due to the photochemical reactions involving NO, $NO_2$, VOCs, and sunlight (Crutzen, 1979, Banta et al., 1998, 2005; Valente, 1998; Langford et al., 2010a; Senff et al., 2010). The exposure to $O_3$ levels higher

than the background can result in damage to biological tissue in crops and living organisms (Haagen-Smit, 1952), and decrease the rate of photosynthesis in plants (Morgan et al., 2003).

The airborne lidar measurements were carried out to assess whether a substantial amount of ozone was generated from the oil sands industrial emissions. The field campaign with the lidar on board a

Twin Otter aircraft consisted of five flights out of Fort McMurray during the period between August 22 and August 26, 2013. The ozone mixing ratio was measured in the unpolluted air upwind, in the polluted air directly above the oil sands operations, and as far as 150 km downwind. The only observed enhancement in ozone occurred in a layer above the polluted boundary layer and air trajectory analysis linked this to forest fire emissions. The measurement technique, analysis methods, results, and

interpretation are described in the following sections.



## 2 Measurement Technique

The differential absorption lidar instrument shown in Figs. 1 and 2 was installed on a Twin Otter aircraft and viewed downward for vertical profile measurements of ozone and aerosol. The lidar transmitter consisted of a Q-switched Nd:YAG laser with second and fourth harmonic generation for

emitting pulsed light with wavelengths of 532 nm and 266 nm at a repetition rate of 20Hz. In Fig. 1, the light with a wavelength of 266 nm was focused into a cell filled with $CO_2$, at a pressure of 965 kPa, to generate light at wavelengths of 276.2 nm, 287.2 nm, and 299.1 nm by stimulated Raman scattering (Nakazato et al., 2007). The UV wavelengths were directed into the atmosphere along the nadir from the Twin Otter aircraft. The laser light at a wavelength of 532 nm was directed independently into the

atmosphere along the nadir. A 15 cm diameter off axis parabolic mirror with a focal length of 500 mm was used to collect the light that was scattered back from molecules and aerosol particles. Two separate optical fibers were positioned behind field stop apertures with diameters of 0.5 mm and 1.0 mm that determine fields of view of 1.0 mrad for the 532 nm and 2.0 mrad for the UV backscatter signals. The four UV wavelengths (266, 276.2, 287.2, and 299.1 nm) were separated in the receiver using the

transmittance and reflectance from interference filters having a bandwidth of 1 nm and tilted at an angle of 7.5 degrees. Photomultiplier tubes (PMT) were used to detect the backscattered light. Photon counting was used to record the weak signals from distances greater than 1.5 km and analog to digital conversion was used for strong signals in the near range. The vertical range bin was 3.75 m for the 532 nm signal and 7.5 m for the UV signals, while the in-flight temporal averaging was 10 seconds for both

the UV and 532 nm signals. During analysis the raw signals were averaged with vertical boxcar smoothing over 23 m for the 532 nm signal and 45 m for the UV signals.

An in-flight visualization tool was used for preliminary real time data analysis and this informed decisions on the flight track. The program provided contour plots of the 532 nm aerosol scattering ratio

and the $O_3$ mixing ratio as measurements were being collected. Line plots of aerosol backscatter and $O_3$ mixing ratio were displayed at the location the cursor was placed on the contour plot for profile-by-profile analysis during the flight.

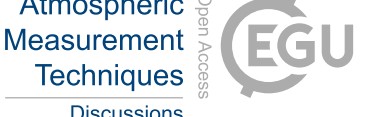



## 3 Analysis of Lidar Measurements

The backscattered optical power as a function of range $z$ is described by the lidar equation as

$$P_\lambda(z) = \frac{C}{z^2} \; \beta(\lambda, z) \; exp\left[-2\int_0^z \left(\sigma_\lambda \, N_{O_3}(z) + \alpha(\lambda, z)\right) dz\right] \tag{1}$$

The constant $C$ includes the emitted pulse energy, sampling interval, telescope area, optical throughput, and detection efficiency. The backscatter coefficient, $\beta(\lambda, z)$, is the fraction of the laser light pulse scattered back to the receiver per unit length through the atmosphere and per unit solid angle. The exponential term represents the round trip transmittance of the laser pulse between the lidar and range $z$.

The extinction coefficient, $\alpha(\lambda, z)$, is the fractional decrease in the laser pulse intensity per unit length in the atmosphere due to scattering. For this experiment, the molecule under investigation is $O_3$ with an absorption cross section of $\sigma_\lambda$. Temperature-dependent $O_3$ absorption cross sections were obtained from the HITRAN 2012 database (Rothman et al., 2013). The number density of $O_3$ molecules in the atmosphere is represented by $N_{O_3}(z)$ in Eq. (1) and the product $\sigma_\lambda \, N_{O_3}(z)$ represents the fractional

decrease in the laser pulse intensity per unit length due to the absorption by $O_3$.

The differential absorption lidar technique was used to determine the mixing ratio of $O_3$. It is based on the difference in the absorption cross section between two laser wavelengths. The measurement wavelength with larger ozone absorption cross section is referred to as the ON

wavelength, $\lambda_{ON}$, and the wavelength with smaller absorption cross section as the OFF wavelength, $\lambda_{OFF}$. Starting from the ratio of separate lidar equations for the ON and OFF wavelengths, the number density of ozone can be extracted by taking the natural logarithm and differentiating with height

$$\qquad\qquad\qquad\qquad [A] \qquad\qquad\qquad\qquad [B] \qquad\qquad\qquad [C]$$

$$N_{O_3}(z) = \frac{-1}{2(\sigma_{O_3,ON} - \sigma_{O_3,OFF})} \left[\frac{d}{dz}\left(ln\left(\frac{P_{\lambda_{ON}}(z)}{P_{\lambda_{OFF}}(z)}\right)\right) - \frac{d}{dz}\left(ln\left(\frac{\beta(\lambda_{ON}, z)}{\beta(\lambda_{OFF}, z)}\right)\right)\right] - \left[\frac{\alpha(\lambda_{ON}, z) - \alpha(\lambda_{OFF}, z)}{(\sigma_{O_3,ON} - \sigma_{O_3,OFF})}\right] -$$

$$\left[\frac{(\sigma_{SO_2,ON} - \sigma_{SO_2,OFF})}{(\sigma_{O_3,ON} - \sigma_{O_3,OFF})}\right] N_{SO_2}(z) \tag{2}$$

$$\qquad\qquad [D]$$



Term A in Eq. (2) represents the $O_3$ number density calculated directly from the lidar signals. The terms B and C in Eq. (2) are corrections for the wavelength dependence in the backscatter and extinction coefficients and term D is a correction for other gaseous molecules that absorb UV radiation, such as

$SO_2$. The number density of $SO_2$ molecules in the atmosphere is represented in Eq. (2) by $N_{SO_2}(z)$. The $O_3$ mixing ratio can be derived by considering six different ON/OFF wavelength pair combinations from the four UV wavelengths that were transmitted by the lidar.

The recorded signals were averaged with vertical boxcar smoothing over 23 m for the 532 nm

signal and 45 m for the UV signals. In order to reduce the uncertainty in the $O_3$ measurements, temporal boxcar averaging was applied over 1.3 minutes (corresponding to a distance of about 7 km along the path of the aircraft). Complete overlap between the transmitted laser pulses and the field of view of the telescope occurred at distances greater than 300 m and signals recorded nearer the aircraft were not used in the analysis. Measurements within clouds have also not been used for retrieving ozone density.

Figure 3a shows an example of backscatter signals (after averaging) recorded from below the aircraft at wavelengths of 266, 276, and 299 nm and Fig. 3b shows the corresponding $O_3$ mixing ratio (derived $O_3$ number density divided by number density of air). For the ozone measurements presented in this paper, the analog measurements at wavelengths 276 nm and 299 nm were used at distances

within 1.8 km of the aircraft. Photon counting measurements at the wavelengths 266 nm and 299 nm were used within the polluted boundary layer.

### 3.1 Correction for Aerosol Interference in the Lidar Ozone Retrieval

When the amount of aerosol in the atmosphere is insignificant, term B in Eq. (2) is very small and term C is straightforward to calculate since it includes only molecular scattering. When there is a significant

amount of aerosol present, terms B and C in Eq. (2) can result in large uncertainties if the aerosol contributions to the backscatter and extinction coefficients are not accounted for. A method for





correcting the ozone retrieval for the wavelength dependence of aerosol extinction and backscatter was developed for this study.

The aerosol backscatter and extinction coefficients at the UV wavelengths were derived by making use of the lidar signal at a wavelength of 532 nm and in situ measurements of aerosol size distribution obtained on the NRC Convair-580 aircraft that was operated for ECCC as part of the JOSM program. Particles with diameters in the range 0.06 to 1 μm were sampled with the UHSAS instrument (Cai et al., 2008) and larger particles ranging in diameter from 0.3 to 20 μm were measured with the FSSP-300 instrument (Baumgardner et al., 1992). For the purposes of correcting the aerosol interference in the lidar ozone measurements, it was assumed that the composition and size distribution of aerosol particles were consistent throughout the boundary layer.

The steps involved in accounting for aerosol in the derivation of ozone from the lidar measurements are summarized as follows:

*A) Aerosol optical properties derived from the 523 nm lidar backscatter signal.*

The profile of extinction and backscatter coefficients was derived from the recorded lidar backscatter signal at the wavelength of 532 nm. The absorption by ozone is not significant at this wavelength so the backscatter and extinction coefficients can be derived from the data independent from the ozone measurement. The method of Fernald (1984) was employed and this required a reference value for the extinction at a particular height and also the ratio of extinction to backscatter (the lidar ratio). The aerosol extinction and backscatter coefficients at the reference height were estimated by using in situ measurements of particle size distribution, *N(r)*, during a Convair flight on August 23, 2013 above the boundary layer (above height 1.5 km ASL) and integrating over all particle sizes as follows:

$$\alpha_{aerosol}(\lambda_{532}) = \int_0^\infty \pi r^2 \, Q_{ext}(m, r, \lambda_{532}) \, N(r) \, dr \tag{3}$$

$$\beta_{aerosol}(\lambda_{532}) = \int_0^\infty \pi r^2 \, Q_{back}(m, r, \lambda_{532}) \, N(r) \, dr \tag{4}$$





The aerosol extinction and backscatter efficiencies ($Q_{ext}$ and $Q_{back}$) were determined from the in situ size distribution measurements with calculations based on the theory of Mie scattering for spherical particles (Bohren and Huffman, 1983). A software module was used to evaluate Maxwell's equations for scattering and absorption of light by homogenous spherical particles. The calculation required a value of

the complex refractive index of particles at the measurement wavelength in order to compute the extinction and backscatter efficiencies. In the region directly above the oil sands (where the maximum aerosol loading occurred), the particle refractive index corresponding to the mineral Kaolinite was used. Kaolinite is a clay mineral composed of aluminum silicate. Studies done by Cloutis et al., (1995), Omotoso and Mikula, (2004), and Mercier et al., (2008), have found kaolinite to be the prominent clay

particle in the oil sands region. The refractive index of kaolinite at a wavelength of 532 nm is 1.57+0.006$i$ (Arakawa et al., 1997).

The in situ particle size distribution measurements acquired directly over the oil sands region within the boundary layer, at an altitude of 785 m ASL, were used to calculate a value of the lidar ratio

(at a wavelength of 532 nm). The lidar ratio was used in Fernald's algorithm to derive the extinction and backscatter coefficient height profiles. Table 1 outlines the input parameters that were used in the Mie scattering calculations. The lidar ratio at a wavelength of 532 nm was found to be 31 sr within the surface boundary layer directly over the oil sands industry. For the aerosol correction in the ozone retrieval it was assumed that this lidar ratio is independent of height. This assumption means that

changes in the optical extinction and backscatter coefficients are attributed to variation in aerosol concentration and not particle size.

*B) Height profile of aerosol properties from combined lidar and in situ measurements.*

The lidar 532 nm aerosol extinction coefficient was combined with the in situ measurements of aerosol

properties to derive the height profile of aerosol number density as follows:

$$N_{aerosol}(z) = \frac{\alpha_{aerosol}(\lambda_{532},z)}{\pi R_{eff}^2\, Q_{ext}(m,R_{eff},\lambda_{532})} \tag{5}$$

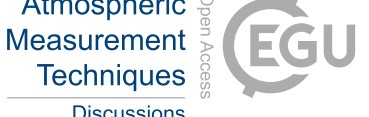

The effective radius of the particles, $R_{eff}$, was found by using the mean cross-sectional area of the particle size distribution:

$$R_{eff}^2 = \frac{\int_0^\infty N(r)\, r^2\, dr}{N_0} \tag{6}$$

$N(r)$ is the number density for particles with radius in the range between $r$ and $r + dr$, and $N_0$ represents the total particle number density. The backscatter and extinction coefficients as a function of height at the UV wavelengths can be derived as

$$\alpha_{aerosol}(\lambda_{UV}, z) = \frac{N_{aerosol}(z)}{N_0} \int_0^\infty \pi r^2\, Q_{ext}(m, r, \lambda_{UV}) N(r)\, dr \tag{7}$$

$$\beta_{aerosol}(\lambda_{UV}, z) = \frac{N_{aerosol}(z)}{N_0} \int_0^\infty \pi r^2\, Q_{back}(m, r, \lambda_{UV}) N(r)\, dr \tag{8}$$

It was assumed that the shape of the aerosol size distribution from the in situ measurement is applicable
throughout the boundary layer. The ratio $N_{aerosol}(z)/N_o$ provides a scaling factor that accounts for changes in the total number density.

*C) Aerosol correction in the ozone retrieval*

The calculated aerosol extinction and backscatter coefficients at the UV wavelengths were substituted
into terms B and C in Eq. (2). Figures 4b, 4c, and 4d show the computed values for the first three terms in Eq. (2) for the measurements taken on August 23, 2013 directly over the oil sands industry where significant aerosol was observed within the boundary layer. In Fig. 4c, the aerosol correction significantly affects term B in Eq. (2) and this term has the greatest contribution to the aerosol correction. The gradient in the aerosol backscatter profile at the top of the boundary layer (between 1.0
and 1.5 km ASL in Fig. 4a) resulted in a correction of up to 13 ppbv in the $O_3$ retrieval. In regions where there is a small gradient in the amount of aerosol (altitudes below 0.8 km in Fig. 4a) the contribution of term B in Eq. (2) is much less.



The magnitude of the aerosol correction in the retrieved ozone mixing ratio (Fig. 4) was consistent with what has been reported from previous ozone lidar studies that employ different methods for calculating the aerosol extinction and backscatter coefficients at UV wavelengths (Alvarez et al., 1998; Eisele & Trickl, 2005). The aerosol correction in previous lidar ozone retrievals was small (< 3 ppbv) in regions of low aerosol loading or in regions where the vertical gradient in the backscatter coefficient was small (Sullivan et al., 2014), but corrections between 15 and 35 ppbv have been calculated due to the presence of large aerosol gradients at the top of the boundary layer (Alvarez et al., 1998; Eisele and Trickl, 2005).

Forest fire smoke was observed in the lidar measurements for the flight on August 24, 2013 and the results are shown in Sect. 5.2. The correction in the lidar ozone retrieval due to the interference of forest fire aerosol was carried out in a similar manner to that described above. In situ particle size distribution measurements in a layer of forest fire smoke were acquired on the Convair-580 aircraft during an ascent above the boundary layer approximately an hour before the time that the lidar observed the layer of smoke at the same height. The complex refractive index of forest fire aerosol that was used in this correction method was taken from Wandinger et al. (2002) and the value is listed in Table 1. The lidar ratio at a wavelength of 532 nm was found to be 65 sr from Mie scattering calculations with the in situ particle size distribution and refractive index for smoke particles. It was found that the correction due to the interference of forest fire aerosol decreased the ozone measurement by a maximum of 7 ppbv. There was some uncertainty since the in situ measurements used in the correction method were taken one hour before the time the lidar observed the layer of forest fire smoke. Different aerosol size distributions from previous studies of biomass burning (Chakrabarty et al., 2006; Pirjola et al., 2015) were used to assess the uncertainty in the amount of correction in the ozone retrieval due to the forest fire smoke. The amount of correction in the ozone measurement when using the size distribution of forest fire smoke found in the previous literature was 5 ppbv (for an average effective radius of 0.09 µm) which is only 2 ppbv smaller than the value of 7 ppbv that was calculated using the in situ size distribution measurements. The amount of correction in the ozone retrieval due to interference of forest



fire aerosol was smaller than the amount of correction due to the interference of oil sands aerosol since the gradients in the amount of forest fire aerosol were smaller.

The term D in Eq. (2) represents the interference by $SO_2$, which absorbs with varying optical cross section within the range of UV wavelengths used for the ozone retrieval. It was found that the wavelength pair 266/299 can result in an bias in the ozone retrieval of up to 0.2% of the $SO_2$ concentration and the largest interference comes from using the 287/299 wavelength pair, at about 37% of the $SO_2$ concentration. For the measurements presented in this paper, the 266/299 wavelength pair was used to retrieve ozone concentration within the boundary layer since the interference from $SO_2$ was a minimum. In situ $SO_2$ measurements were acquired on the Convair-580 aircraft and mixing ratios typically ranged between 30 and 150 ppbv within the boundary layer. An $SO_2$ concentration of 150 ppbv would result in a 0.3 ppbv change in the ozone mixing ratio derived from the lidar measurements with the 266/299 pair. Corrections for $SO_2$ were not applied to the lidar ozone retrieval since the actual concentration of $SO_2$ along the Twin Otter flight path was not known and the magnitude of the correction was small.

## 3.2 Corrections for Detection Nonlinearity and Signal Induced Noise

For weak signals measured with the PMT, the photon count rate is proportional to the incident optical power. At high count rates (greater than 10 MHz), the photon counting signal does not respond linearly to the optical power due to overlapping pulses from the detector being counted erroneously as a single pulse. The detection system is unable to count separate individual pulses accurately within a certain time period commonly referred to as dead time $t_d$ (Donovan et al., 1993). A dead time correction was applied to the raw photon counting data by calculating the true count rate $N_T$, in terms of the measured count rate $N_m$, as $N_T = N_m/(1 - N_m \times t_d)$. The true count rate was found by using a value of a dead time for which the ratio of $N_T$ to the recorded analog signal is constant up to a count rate of 100 MHz. Unique photon counting dead times were determined for each of the five individual photomultiplier detectors and the values ranged from 4.5 to 7.5 nsec. The optimal dead time correction at each wavelength was determined from lidar measurements acquired along one leg of the Twin Otter flight on



August 22, 2013 and these dead time corrections were applied to the measurements collected on all subsequent flights. The value of dead time correction at each wavelength was consistent throughout the campaign for each PMT.

5       Another correction was applied to the UV signals in order to remove a residual decaying signal in the far range. Strong UV signals in the near range can introduce a residual decaying signal in the far range, most commonly referred to as the signal induced noise or signal induced background. The cause of this signal induced noise/background is likely UV fluorescence from the PMT (Zhao, 1999) and this occurs only in the UV wavelength range. This residual signal had an amplitude that was proportional to

the relatively large signal amplitude at near range. It can be modeled by an exponential function (Sunesson et al., 1994). The residual signal was corrected by fitting to an exponential decay function to the signal recorded at distances beyond the ground, where there would be no real optical signal. The exponential fit was then subtracted from the lidar backscatter signal.

**4 Air Trajectory Calculations**

The Hybrid Single Particle Integrated Trajectory (HYSPLIT) model (Draxler & Hess, 1998) was used to predict the trajectory of the emissions released from the oil sands in both the forward and backward directions. Forward air trajectories from the oil sands locations were used for flight planning and backward air trajectory calculations were used in the analysis to reconstruct the past motion of air parcels. The HYSPLIT model was accessed through the NOAA ARL READY website

(http://www.arl.noaa.gov/HYSPLIT.php) and the trajectories were computed by using the Global Data Assimilation System (GDAS) meteorological dataset. The uncertainty in the trajectory calculation from the HYSPLIT model has been assessed to be less than 20% of the travel distance over a travel time of 48 hours (Baumann & Stohl, 1997). In this paper, air trajectories were used in the interpretation of the lidar measurements. The largest travel time for a backward trajectory used in the analysis was 10 hours.

The uncertainty in the air trajectory related to a travel time of 10 hours did not affect the final interpretation or conclusions derived from using these trajectories.



## 5 Observations

The Twin Otter flight segments were straight and usually oriented either parallel or perpendicular to the wind direction. The goal was to obtain measurements in regions upwind and downwind from the oil sands pollution emission sources. The path of the Twin Otter aircraft for each of the 5 flights during the campaign is shown in Fig. 5. The wind direction used for flight planning was from a forecast, but the wind direction indicated on each map was obtained from the back trajectories (GDAS dataset).

### 5.1 Industrial Pollution

A typical flight leg from August 23, 2013 that covered areas upwind, above, and downwind from the oil sands industry is shown in Fig. 6. The height of the aircraft was 2.95 km above sea level (ASL). The aerosol optical extinction coefficient and the ozone mixing ratio derived from the lidar measurements along this flight leg are shown in Fig. 7. Measurements collected upwind of the industry (within 20 km from the start of the flight leg) in Fig. 7a show small amounts of background aerosol present in this region. Directly above the oil sands industry (distances of 35 to 65 km along the flight track) significant amounts of aerosol were observed to be mixed to a height of 1.5 km ASL (1 km above ground). Downwind of the industry (distances of 90 to 120 km along the flight leg) the aerosol was dispersed to heights of up to 2.5 km ASL (or 1.8 km above ground). The depth of the boundary layer was apparent from the vertical range over which the aerosol was mixed. This varied from 1.2 to 1.5 km above sea level (or 0.7 to 1.0 km above ground) directly above the industry. A flue-gas desulfurization (FGD) stack was intersected at 57° N and 111°39' W in Fig. 6. The vertical plume from the stack is clearly seen in the lidar aerosol measurement in Fig. 7a as the presence of a vertical extension in the depth of the aerosol layer at the distances of 55 to 60 km from the start of the flight leg.

Figure 7b shows the corresponding ozone mixing ratio along the flight segment. Regions of reduced ozone mixing ratios between 15 and 34 ppbv were measured in the polluted boundary layer directly above the oil sands industry (distances from 30 km to 65 km along the flight leg). The ozone mixing ratio upwind and downwind from the industry varied between 25 and 40 ppbv.





A case is presented in Figs. 8 and 9 with a segment of the flight on August 22, 2013 that was oriented transverse to the path of the air that passed over the oil sands industry, approximately 100 km downwind. The flight segment was south along longitude 110˚ W between positions I and J (Fig. 8), and the corresponding lidar measurements are shown in Fig. 9. The optical extinction coefficient

derived from the lidar measurements (Fig. 9a) indicated that the depth over which the aerosol was mixed from the ground ranged from 1.5 to 2 km ASL (1.0 to 1.5 km above ground). The $O_3$ mixing ratio within the boundary layer varied between 27 and 38 ppbv throughout the flight segment. Ozone mixing ratios greater than the background were not observed.

Backward air trajectories were calculated from locations along all of the Twin Otter flight tracks during the campaign. For each measurement point along the flight tracks, the time was determined for the air to travel between the oil sands industry and the position of the lidar measurement from the Twin Otter. Only the trajectories that passed over the oil sands industry were selected for this analysis. Figure 10 shows lidar measurements of the aerosol optical extinction coefficient and $O_3$ mixing ratio as a

function of time taken by the air to travel between the oil sands industry and the measurement point. The measurements were averaged within the height range of 500 m to 800 m ASL). As expected, Fig. 10a shows that larger values of aerosol optical extinction coefficient occur closer to the pollution source area and decrease moving away from the source. The observed $O_3$ mixing ratio in Fig. 10b showed no trend with time or distance from the industry. There is no evidence for increasing $O_3$ for times since

passing over the industry of up to 10 hours. The only deviation from the background were instances of smaller ozone mixing ratios.

### 5.2 Forest Fire Emissions

A case with air pollution from both natural and industrial sources is shown in this section. Airborne lidar measurements to the north of Fort McMurray were made on August 24, 2013 and the

corresponding flight path is shown in Fig. 11. In this flight leg, the Twin Otter started at point A, a distance of ~90 km to the east of the oil sands industry and travelled westbound to point B along constant latitude of 57°06' N.





In the contour plot of optical extinction coefficient (Fig. 12a) there was a layer of aerosol in the height range 1.5 to 2.5 km ASL at distances of 15 km to 90 km from point B, and this layer was separated from the industrial pollution in the surface boundary layer. The aerosol from the oil sands

industrial emissions was confined within the depth of the boundary layer, below a height of 1.2 km ASL at distances of 20 km to 55 km from point B. It was observed visually from the aircraft that the separated aerosol layer above the boundary layer originated from forest fires to the southwest. This was consistent with the approximate locations of forest fires provided from the Alberta Forestry and Emergency Response Division as indicated in Fig. 11. A backward trajectory is shown in Fig 11 that

was initiated from an altitude of 2.0 km at the time and location of the lidar measurement of the aerosol layer that was observed to be separated above the boundary layer. The air containing the separated aerosol layer had passed over the vicinity of the forest fires southwest of the flight track. This aerosol layer is considered to be forest fire smoke.

The $O_3$ mixing ratio measured with the lidar along this flight segment is shown in Fig. 12b. The $O_3$ mixing ratio reached a maximum of 70 ppbv at a height of 2 km, within the separated aerosol layer that originated from the region of forest fires (distances 15 km to 30 km from point B). In the pollution from the oil sands industry (below the forest fire smoke layer), significant amounts of aerosol were observed in which the $O_3$ mixing ratio varied between 15 and 33 ppbv. In the eastern half of the flight

leg the $O_3$ mixing ratios were consistent with background values (25 – 36 ppbv).

### 5.2.1 Ground Based Lidar Polarization Measurements

A ground based lidar was operated by Environment and Climate Change Canada (ECCC) at the location AMS 13, which was approximately 5 km north of the Twin Otter flight track as indicated in Fig. 11. The ECCC lidar observed a distribution of aerosol that was similar to the airborne lidar measurements

in Fig. 12a. It detected the same layer of forest fire smoke in the height range of 1.5 to 2.5 km ASL. The ECCC lidar had an additional capability for measuring the polarization in the backscatter signal. The ratio of the perpendicular to parallel components of polarization in the aerosol backscatter signals (the



polarization ratio), provides an approximate method for discriminating between particles of different sizes and shapes. For example, hexagonal ice crystals can cause a depolarization ratio greater than 0.5 whereas spherical water droplets result in a small (almost zero) depolarization ratio (Sassen et al., 1991). For this case, the depolarization ratio was used to discriminate between the aerosol from the oil

sands operations and forest fire smoke.

The difference in the depolarization ratio between the industrial pollution and forest fire smoke is clearly seen in Fig. 13. The depolarization ratio throughout the forest fire smoke layer (heights 1.5 km to 2.3 km) was measured to be 5-6% and this value is consistent with previous depolarization lidar

measurements of forest fire aerosol (Mattis et al., 2004; Murayama et al., 2004; Pereira et al., 2014). The small values of the depolarization ratio measured in the smoke layer reveal a more spherical habit in the shape of forest fire particles. The depolarization ratio in the pollution from the oil and gas extraction facilities at altitudes below 1.2 km ASL had a larger depolarization ratio with values between 7% and 10%. The larger depolarization ratio observed in industrial pollution suggests that the particles

are less spherical in nature and more likely to be categorized as mineral dust. This provides further evidence that the separated layer at heights of 1.5 km to 2.5 km with the enhanced $O_3$ mixing ratio had originated from forest fires rather than industrial pollution.

### 5.3 Comparison Between Lidar and In Situ Measurements

The Twin Otter and Convair aircraft were not coordinated to fly along the same flight tracks. The

airborne lidar measurements were taken over long and straight flight segments, while the in situ measurements were concentrating on specific emission sources during the period when both aircraft were operating. So a direct comparison between the lidar and in situ measurements for the purposes of validation was not straightforward. There were a few cases when the location where the Convair carried out a spiral ascent or descent could be linked to the location of the Twin Otter lidar measurements since

the air trajectory passed through both locations.





Two cases are presented here where vertical profiles of $O_3$ mixing ratio derived from lidar measurements were compared with the in situ $O_3$ measurements that were acquired on the Convair-580 aircraft during spiral ascents on August 23, 2013. Locations are indicated in Fig. 14 as A-in situ and B-in situ (designated by a star-shape symbol) where in situ measurements of the vertical profile of $O_3$ were

collected during two spiral ascents with the Convair-580 aircraft. The Convair spiral ascent at the point labelled as A-in situ in Fig. 14 was carried out in polluted air above the industry. The Twin Otter did not pass directly over this point, but the lidar measurements used for comparison were obtained 2.5 hours later at a location along the Twin Otter flight track where the back trajectory of the air passed over point A-in situ.  The in situ and lidar measurements of the vertical profiles of ozone mixing ratio are shown in

Fig. 15a. The two separate measurements were within the limits of measurement uncertainty throughout the overlapping common height range (0.8 km to 1.2 km), where the $O_3$ mixing ratio was in the range 20–30 ppbv in the polluted air. At point B-Lidar in Fig. 14, the Twin Otter aircraft collected measurements at a distance of 12 km away from the location of a Convair spiral ascent. The air back trajectories did not pass over any pollution sources in the upwind direction for either case and this is

referred to as unpolluted air. The lidar and in situ measured $O_3$ mixing ratios shown in Fig. 15b were within the limits of uncertainty except between heights of 1.0 km and 1.2 km where the difference had a maximum of 5 ppbv.

The statistical distribution of $O_3$ mixing ratio was also used for comparing the lidar and in situ

measurements. Backward trajectories were computed from an altitude of 800 m ASL along all the Twin Otter and Convair flight tracks during the August 22 to August 26 time period. The trajectories were separated into two categories. The air trajectories that did not pass over the oil sands industry were categorized as unpolluted air and the trajectories that passed over the industry were categorized as polluted air. In situ measurements of $O_3$ were acquired at a single height and the lidar measurements

were averaged between heights of 550 m and 900 m for this analysis. Figure 16a shows a histogram of the $O_3$ measurements in unpolluted air for both the lidar and in situ measurements collected within the surface boundary layer. The peak of the distribution for the lidar and in situ measurements were both at approximately 30 ppbv while the wings of the distributions were somewhat offset. In the polluted air



(Fig. 16b) the distributions were similar with the main peak at 25 ppbv for the lidar and in situ measurements, and both also had secondary peaks at 38 and 39 ppbv. In both the vertical profile (Fig. 15) and statistical comparisons (Fig. 16) the lidar and in situ measurements were consistent with each other and with the general result that the ozone mixing ratio was smaller than the background value in the polluted air directly above the oil sands extraction industry.

## 6 Discussion

A general result from the lidar measurements was that the amount of ozone in the polluted air directly above the oil sands operations was smaller than in the background unpolluted air. The in situ measurements of NO, $NO_2$, and $O_3$ from the Convair-580 aircraft were used to investigate the reason for the deceased ozone abundance. A flight segment of the Convair aircraft on August 23, 2013 (between 11:27 and 11:37 AM local time) that travelled eastward at a constant altitude of 650 m ASL while intersecting the oil sands industry is shown in Fig. 17. Figure 18 shows the in situ measurements of $O_3$, NO, and $NO_2$ mixing ratio along the Convair flight segment shown in Fig. 17. In regions upwind and further downwind of the oil sands industry, the amount of NO was measured to be less than 2 ppbv while the ozone mixing ratio ranged between 25 and 30 ppbv. Directly over the oil sands industry (the region in Fig. 18 contained within the dashed lines), the mixing ratio of NO increased to 25 ppbv while the ozone mixing ratio decreased from 30 ppbv to 13 ppbv. The reduction in $O_3$ over the industry is consistent with NO titration: $NO + O_3 \rightarrow NO_2 + O_2$. The sum of the $NO_2$ and $O_3$ mixing ratios in Fig. 18 remained relatively constant, such that the decrease in $O_3$ was approximately compensated by an increase in $NO_2$.

Another way to describe this is in terms of odd oxygen, $O_x$ ($O_3 + NO_2$) (Brown et al., 2006). The $O_x$ mixing ratio (shown in Fig. 18) remained relatively constant along the flight track, with a slight increase within the industrialized portion of the track despite the significant decrease in $O_3$. The decrease in $O_3$ was slightly more than compensated by the increase in $NO_2$, which is entirely consistent with a titration of $O_3$ in a combustion plume where the $NO_x$ composition of the source is mostly NO (e.g., ~90% NO, 10% $NO_2$), with negligible photochemical formation of $O_3$ close to the source.



It was observed that $O_3$ mixing ratios at distances as far as 150 km downwind of the pollution sources, and up to ten hours since emission, were in the range of the regional background levels. There were no measured enhancements in ozone that are usually found in polluted air and this is consistent with previous measurements of ozone in the Fort McMurray oil sands region (Rudolph 2004). This result was different from what has normally been observed in polluted air. For example, previous studies of air pollution surrounding the power plants, refineries, and petrochemical industry near Houston, Texas have observed that $O_3$ mixing ratios greatly exceeded the regional background (Banta et al. 2005, Senff et al., 2010; Langford et al. 2010b). One obvious difference in comparing the Fort McMurray region to Houston is the temperature. The air temperatures measured from the Convair aircraft over the Fort McMurry oil sands region during late August 2013 were less than 20° C. The surface temperatures in the vicinity of Houston reported by Senff et al. (2010) were in the range of 30° C to 42° C. It is well known and documented that episodes of substantial ozone generation due to pollution occur in hot and stagnant conditions (e.g. Banta et al, 1998; Valente et al., 1998; Jacob et al., 1993, 2009; Lin et al., 2001; Camalier et al., 2007; Coates et al., 2016; Shen et al., 2016).

In addition to the relatively cold temperatures, the conditions could not be characterized as stagnant (Camalier et al., 2007) for most of the flights. Figure 19 shows the 24 hour back trajectory for each day of the Twin Otter flight campaign at midday. With the exception of August 25[th], 2017, the air had travelled at least 200 km in the 12 hours prior to passing over the oil sands region. An increase in the aerosol layer depth downwind of the oil sands emissions (e.g. Fig. 7) provided evidence for vertical mixing. As the pollutants mixed with the clean background air, $O_3$ mixing ratios downwind of the industry gradually increased to between 25 and 40 ppbv and were consistent with background levels.

The absence of enhanced ozone in pollution downwind of the industry is interpreted as being a result meteorological conditions that were not favourable for the generation of ozone. These factors include (A) ambient temperatures not greater than 20° C, (B) no regional stagnation of air, and (C) vertical mixing of the polluted air with clean background air.



An enhancement in the ozone abundance was detected in the forest fire emissions that were encountered above the boundary layer (Fig. 12b). It has been established previously that forest fire emissions include the precursors for generation of ozone (e.g. Jaffe and Wigder, 2012). In this case the

environmental conditions were also favourable for generation of ozone. The temperatures would have been greater in the plumes above the fires. The layer remained well defined and separated from the turbulent boundary layer. It was not subjected to ventilation from the boundary layer convection cells, and the mixing with background air was weak enough that ozone generation could proceed in the middle of the layer.

**7 Conclusion**

A lidar instrument for measurements of atmospheric aerosol and ozone was developed for field deployment. It was installed on a Twin Otter aircraft for a flight campaign to study the impact of air pollution from the oil sands extraction industry in northern Alberta. A correction for the interference of aerosol was required for the retrieval of ozone concentration from the UV differential absorption lidar

measurements. An aerosol correction method was developed that made use of in situ measurements of the aerosol size distribution in combination with lidar measurements. It was found that the abundance of ozone in the pollution directly above the oil sands operations was reduced from what was measured in the background air. In situ measurements of NO, $NO_2$, and $O_3$ on board the separate NRC Convair-580 aircraft were used to show that the reduction in ozone abundance was consistent with NO titration. It

was also found that there was no increase in ozone abundance in the industrial pollution as it was transported downwind to distances of 150 km and times of up to 10 hours. The lack of substantial ozone generation was attributed to conditions that were not favourable for the generation of ozone: low temperatures, lack of stagnation, and vertical mixing with clean background air. A layer of forest fire emissions that was separated from the turbulent boundary layer was observed to contain an increase in

ozone abundance to 70 ppbv. This was consistent with the conditions being more favourable for ozone generation within the forest fire emissions: greater temperature and less mixing with the background air.





**Acknowledgements**

Financial support for this study was provided by FedDev Ontario (through Communitech Corp.), Environment and Climate Change Canada (ECCC), the Natural Sciences and Engineering Research Council of Canada (NSERC), and the Canadian Foundation for Innovation (CFI). Data from radiosonde

measurements were obtained from the University of Wyoming, Department of Atmospheric Science website. The authors would like to thank Mr. Cordy Tymstra from the Forestry and Emergency Response Division (Environment and Sustainable Resource Development of Alberta) for providing records of forest fire locations.

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



**Table 1.** Input parameters used in Mie calculations for the aerosol correction in the lidar ozone retrieval.

| Quantity | Oil sands aerosol | Forest fire aerosol |
|---|---|---|
| Particle effective radius, $R_{eff}$ | 0.072 µm | 0.071 µm |
| Lidar ratio, $LR$ (532 nm) | 31 sr | 65 sr |
| Particle refractive index, $m$ | $\lambda_{530nm} = 1.57 + 0.006i$ $\lambda_{260nm} = 1.68 + 0.041i$ | $1.61 + 0.06i$ |







**Figure 1: Schematic diagram of the differential absorption lidar system used for measurement of ozone and aerosol during the oil sands field campaign.**



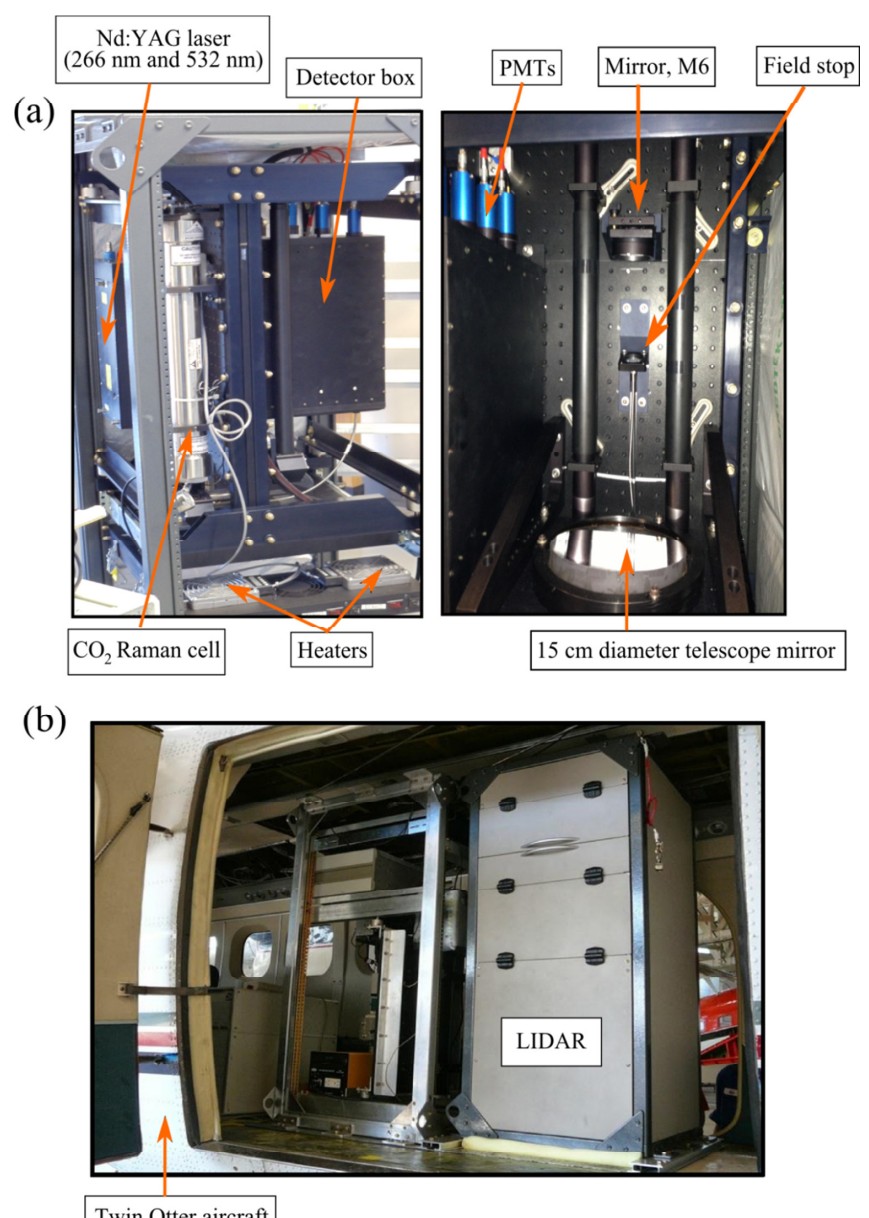

**Figure 2: The lidar system installed (a) in the aircraft rack and (b) on the Twin Otter aircraft.**





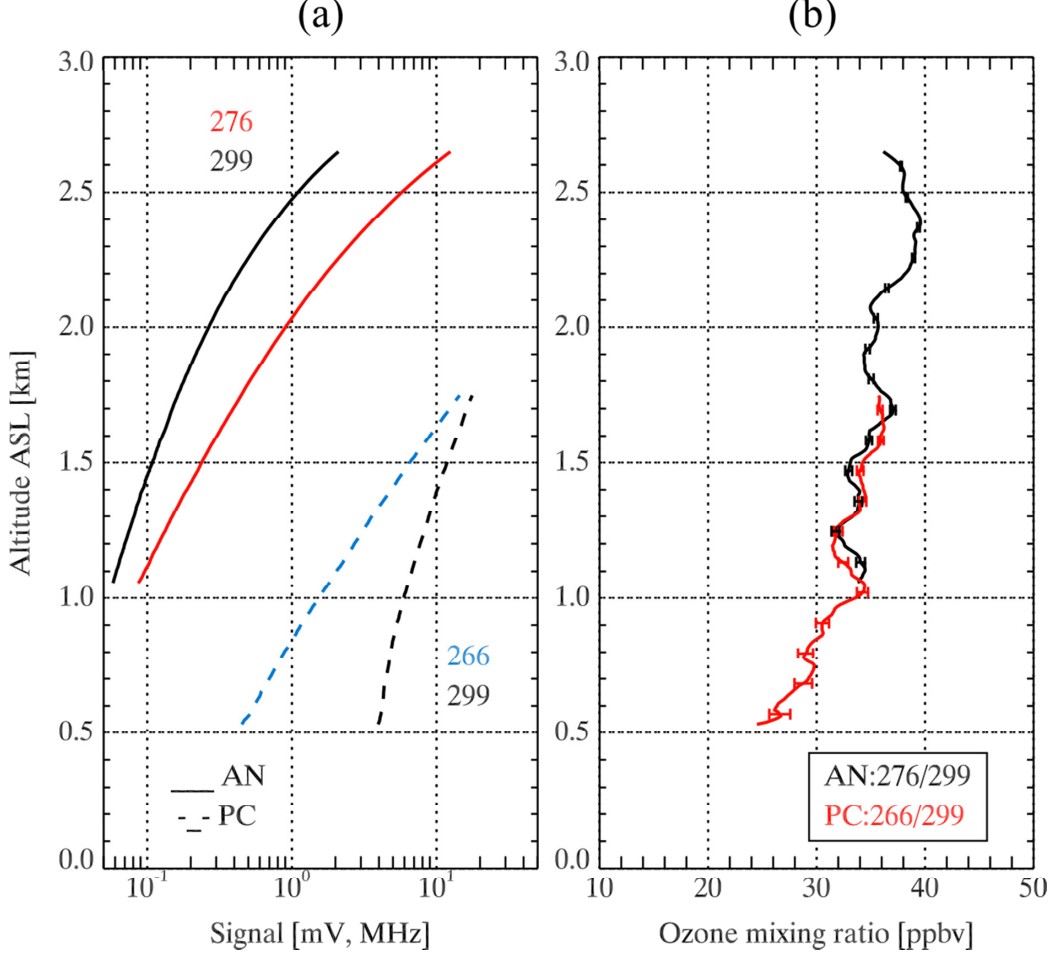

**Figure 3: (a) Lidar backscatter signals for the analog (AN) and photon counting (PC) measurements. (b) The O₃ mixing ratio derived by using the AN:276/299 and PC:266/299**
5 **wavelength pairs. The error bars represent the relative uncertainty corresponding to the standard deviation in the photon counting. The measurements shown here were collected on August 26, 2016 at 5 PM local time (UTC – 6h) as the Twin Otter flew downwind of the oil and gas production facilities.**




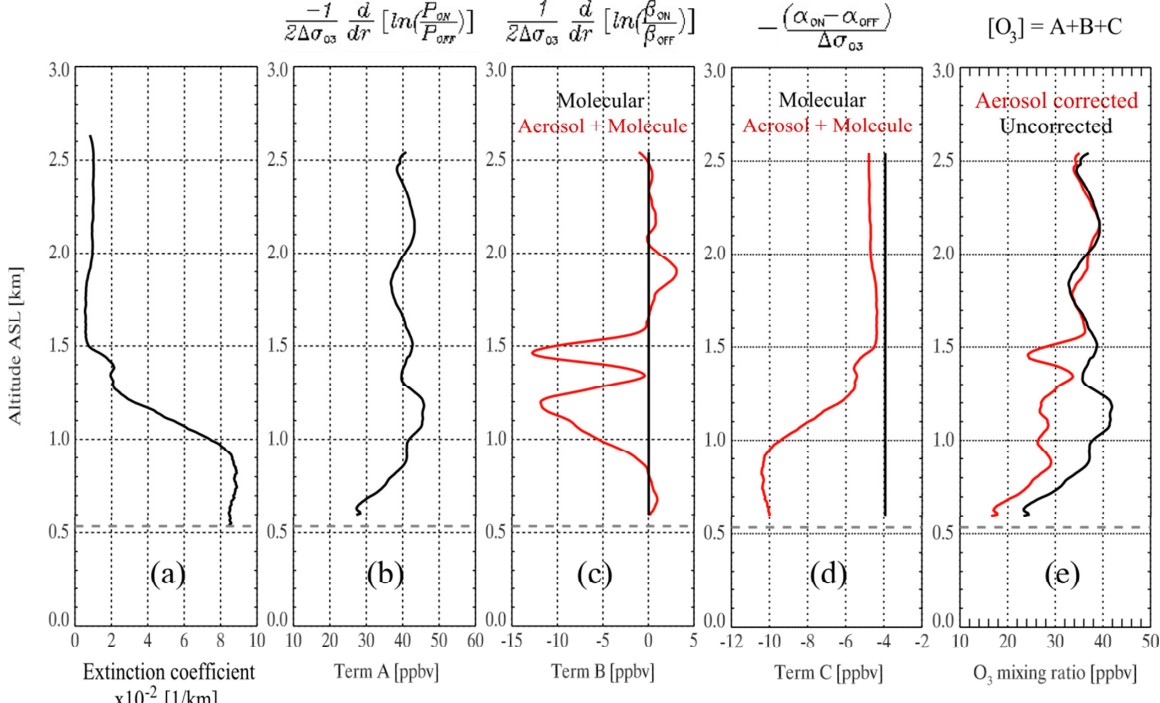

**Figure 4: The O₃ retrieval for the 276/299 wavelength pair. Altitude is above sea level (ASL) (a)
The aerosol optical extinction coefficient derived by using the lidar measurement at a wavelength
of 532 nm. Calculation of term A (panel b), term B (panel c), term C (panel d) from Eq. (2). (e)
The addition of the first three terms in Eq. (2). The line plots in red represent the aerosol
corrected term and the dashed line in all of the plots represents the height of the ground. The
measurements shown here were collected on August 23, 2013 at 1:15 PM local time (UTC – 6) as
the Twin Otter flew directly above the oil sands industry where significant amounts of aerosol
were mixed up to an altitude of 1.5 km ASL.**





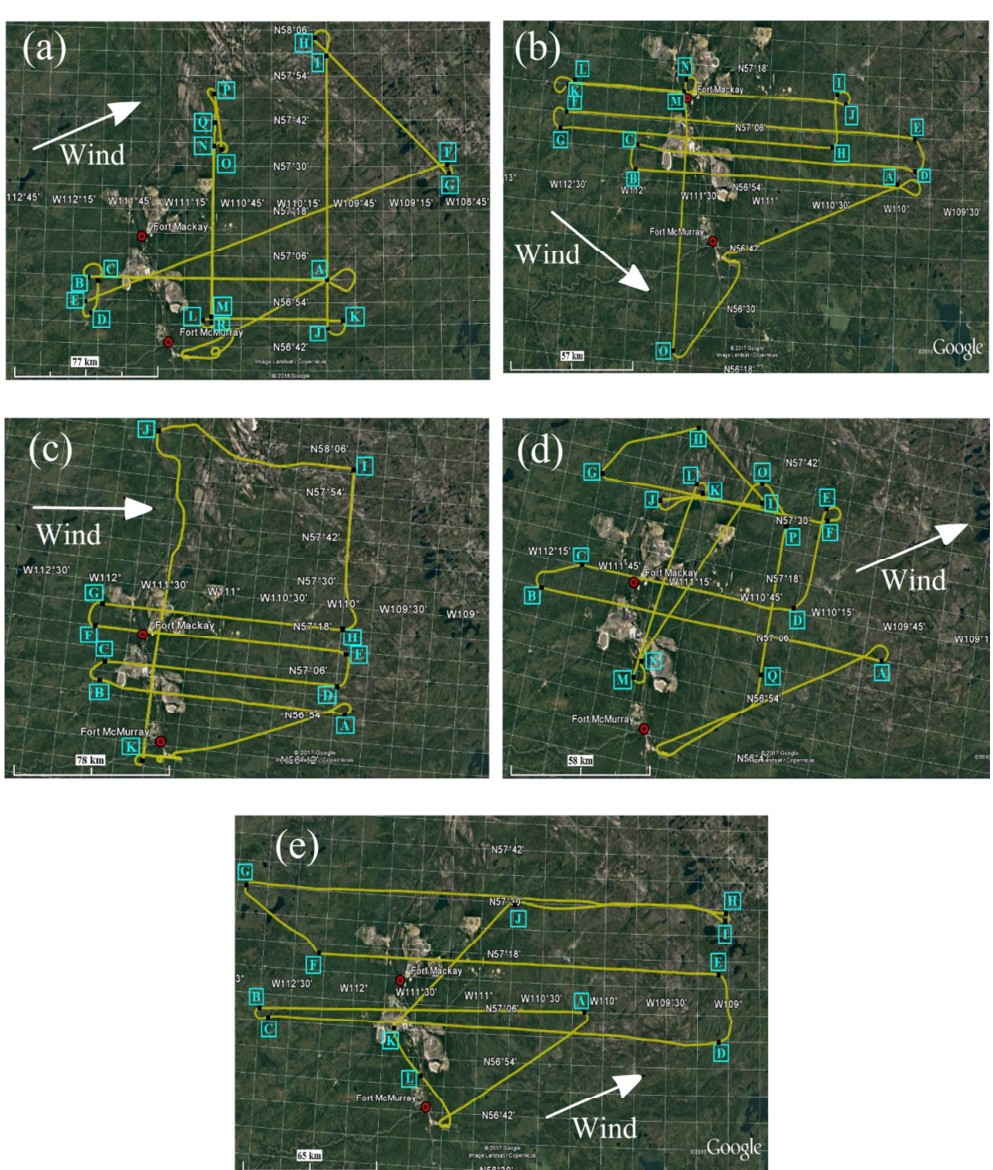

**Figure 5: Flight track of the Twin Otter aircraft on (a) August 22, (b) and (c) August 23, (d) August 24, and (e) August 26, 2013. The direction of wind at the start of each flight is represented by the arrow.**





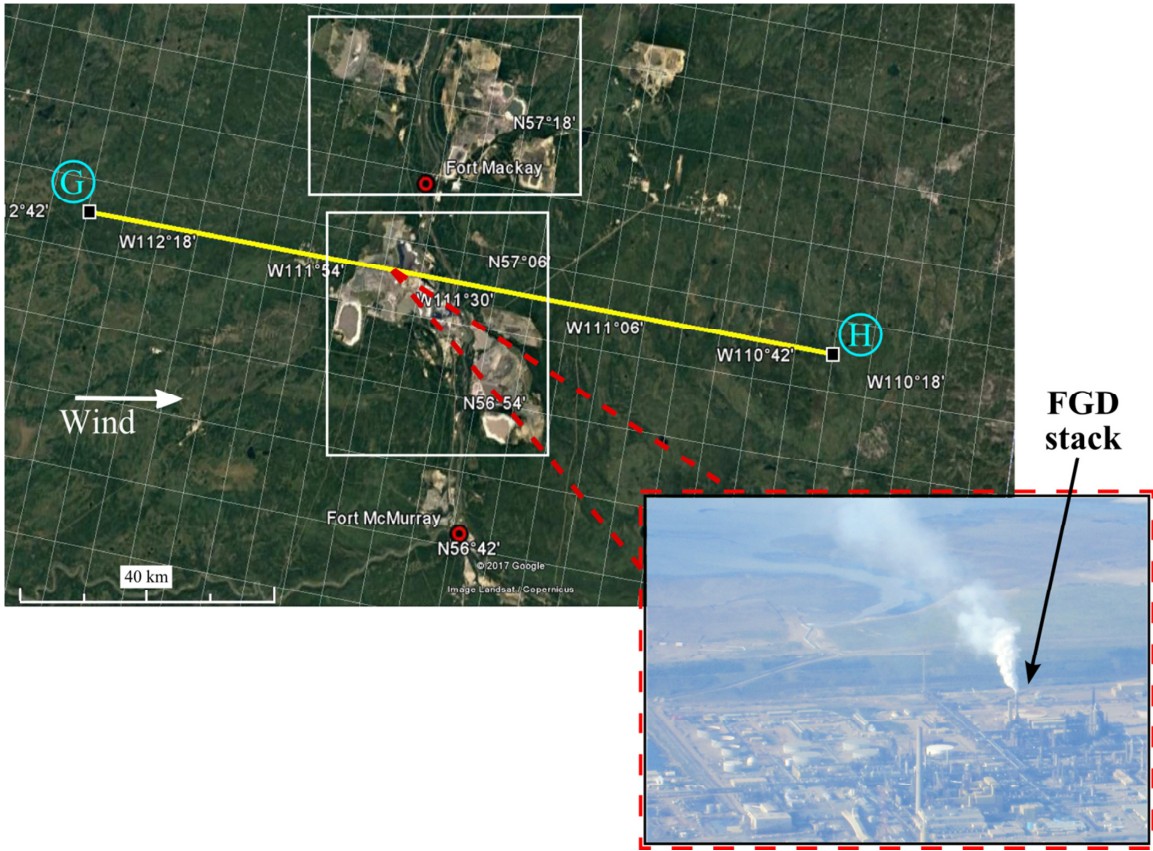

**Figure 6: The flight segment G-H on August 23, 2013 (1:04 to 1:26 PM local time (UTC – 6h)
displayed on GoogleEarth. Point G represents the starting position of the flight leg and point H
the ending position. The oil sands industry was contained within the regions outlined in white.
The average direction of the wind at height of 800 m ASL on August 23, 2013 at the measurement
time is indicated by the arrow. Inset photo shows the area that was flown over, including the flue-
gas desulfurization (FGD) stack.**





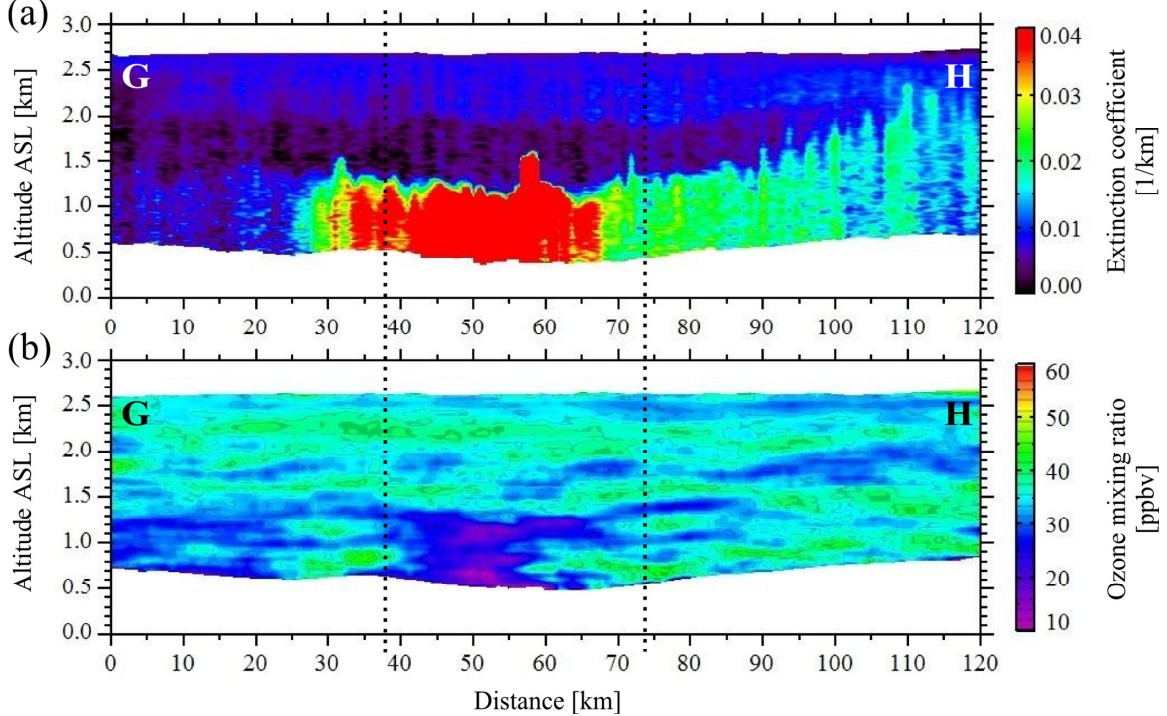

**Figure 7: (a) The aerosol optical extinction coefficient derived from the lidar measurements at a wavelength of 532 nm and (b) the ozone mixing ratio derived from the UV lidar measurements for the flight segment G-H on August 23, 2013. The lidar measurements were collected between 1:04 and 1:26 PM local time (UTC – 6h). The height is above sea level (ASL) and the distance is along the flight segment in Fig. 6. The measurements contained within the vertical dashed lines represent the part of the flight segment directly above the oil sands contained within the regions outlined in white in Fig. 6.**





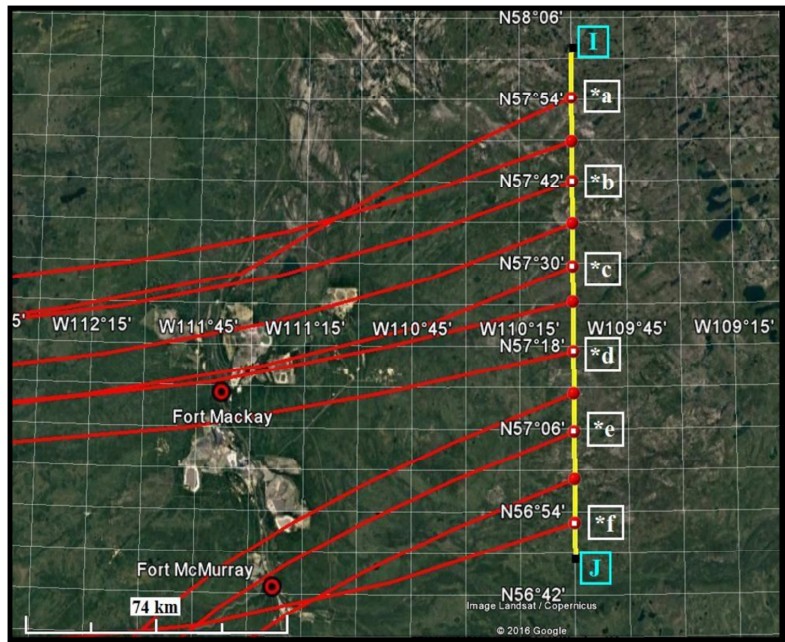

**Figure 8: The flight segment I-J on August 22, 2013 is represented by the yellow line. Point "I" represents the starting position of the flight leg and point "J", the ending position. Backward air trajectories initiated from an altitude of 1000 m above sea level on August 22, 2013 at 2:00 PM local time (UTC – 6h) are shown in red and marked by the points \*a to \*f.**

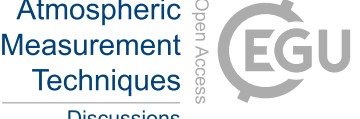

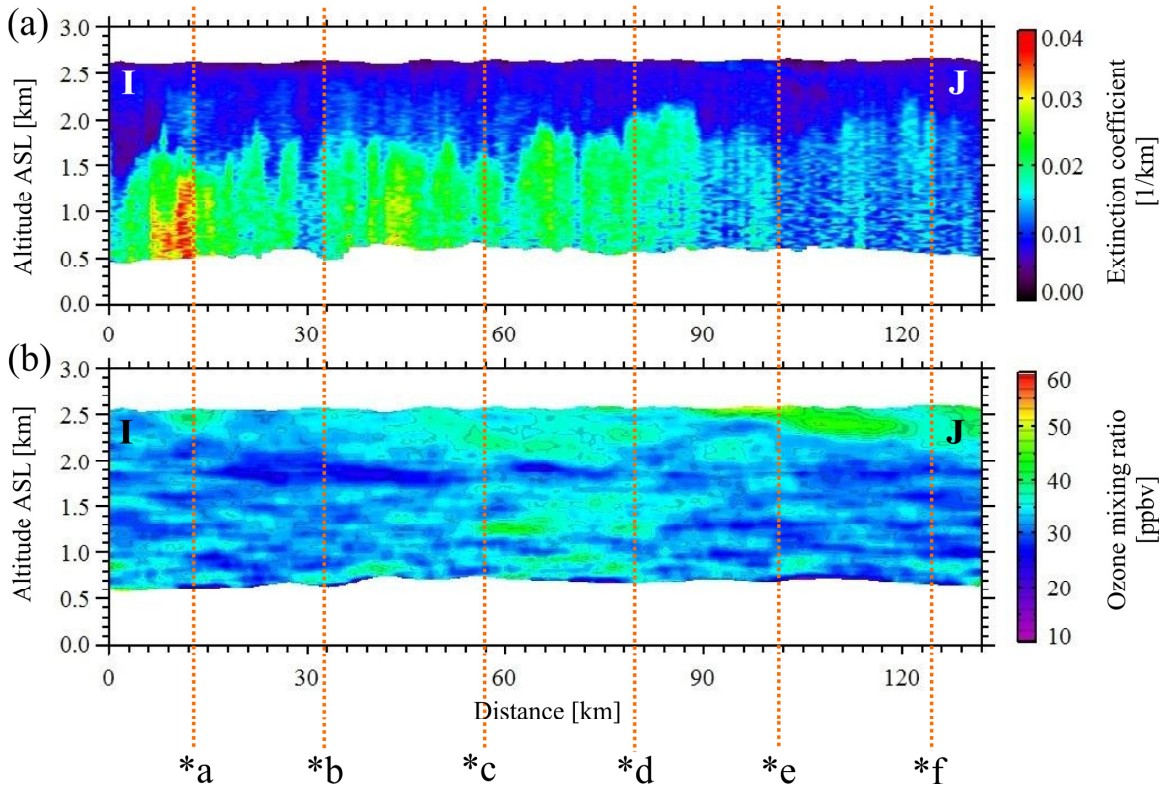

**Figure 9: (a) The aerosol optical extinction coefficient measured at a wavelength of 532 nm and (b) the ozone mixing ratio derived from the UV lidar measurements for the flight segment I-J on August 22, 2013. The lidar measurements were collected between 1:52 and 2:20 PM local time (UTC – 6h). Points \*a to \*f represent the location of backward trajectories along the measurement path and are marked on the flight segment in Fig. 8.**





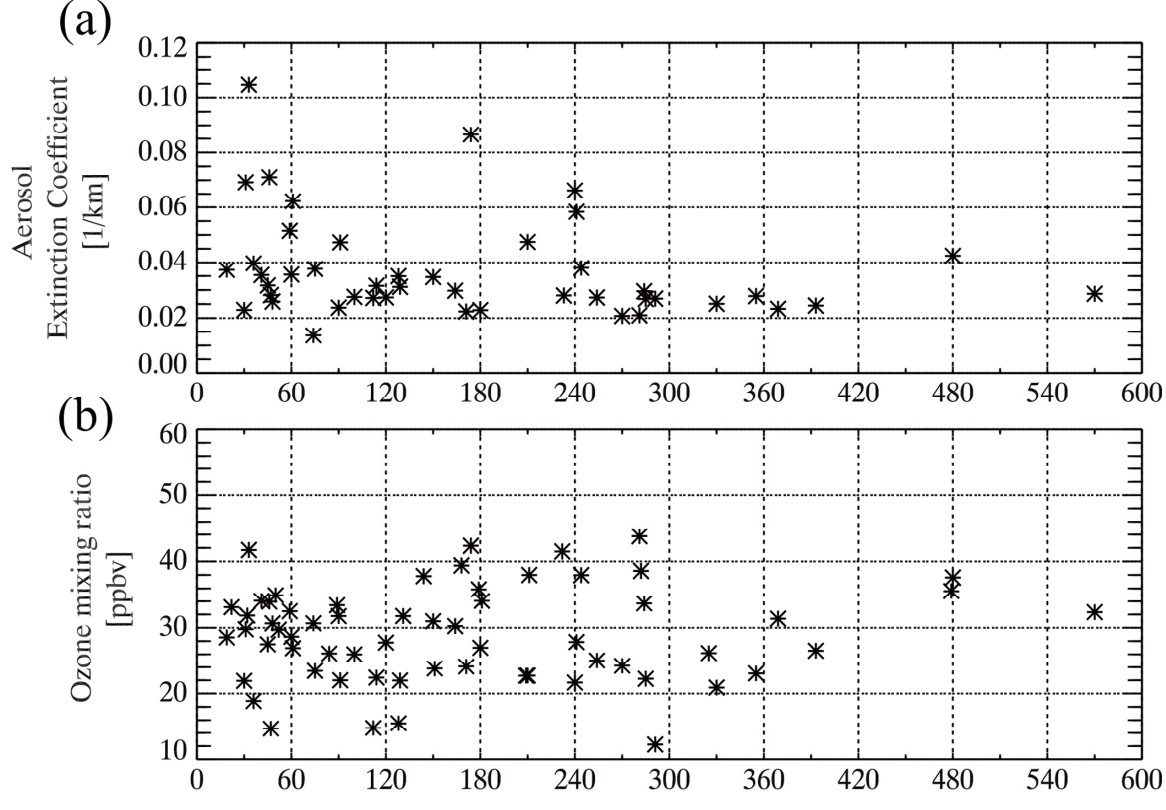

**Figure 10: (a) The aerosol optical extinction coefficient and (b) the ozone mixing ratio as a function of time since the air had passed over the industry to reach the measurement point along all Twin Otter flight tracks. The measurements shown here were collected between August 22 and 26, 2013.**



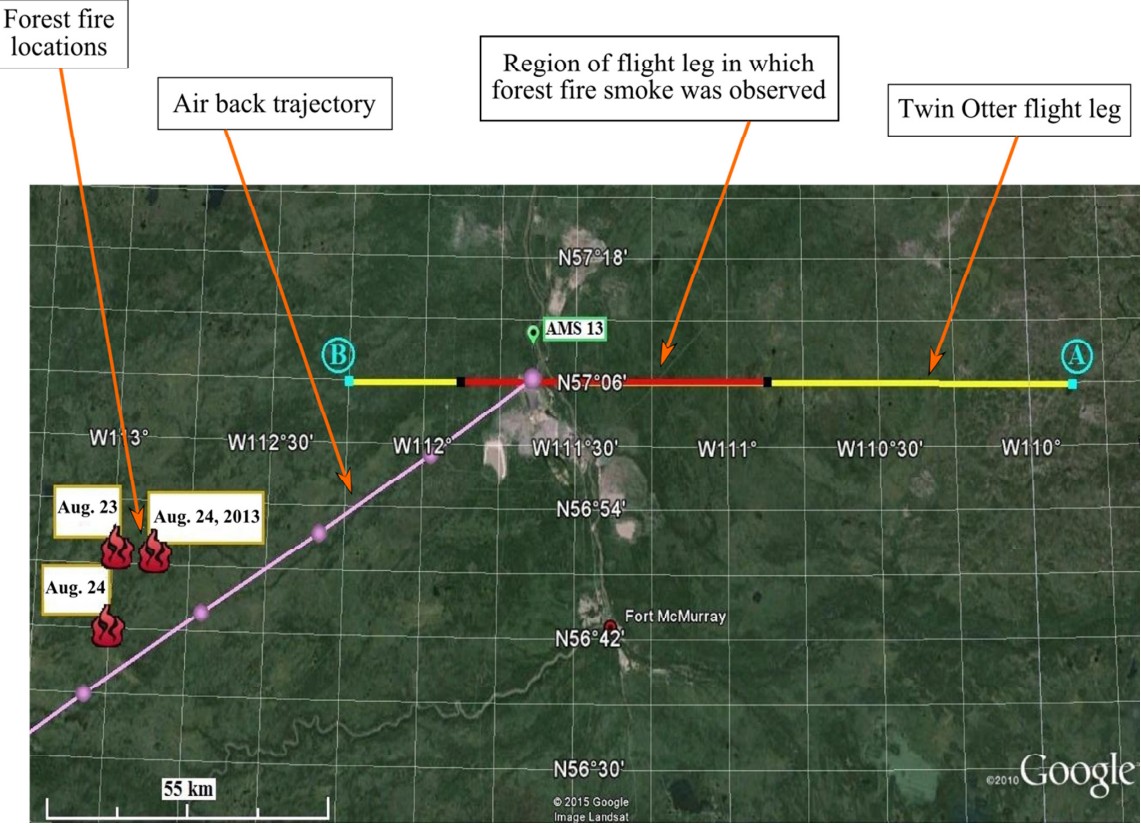

**Figure 11: The flight segment A-B on August 24, 2013. The lidar measurements were collected between 2:48 and 3:18 PM local time (UTC – 6h). The red section along A-B represents the region where forest fire smoke was observed. The forest fires are depicted by red fire symbols. A backward air trajectory was initiated from an altitude of 2.0 km ASL at a time of 3:00 PM local time on August 24, 2013. The round marks along the trajectory represent a time interval of 1 hour. The location of the ECCC ground based lidar is indicated as AMS 13.**

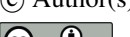



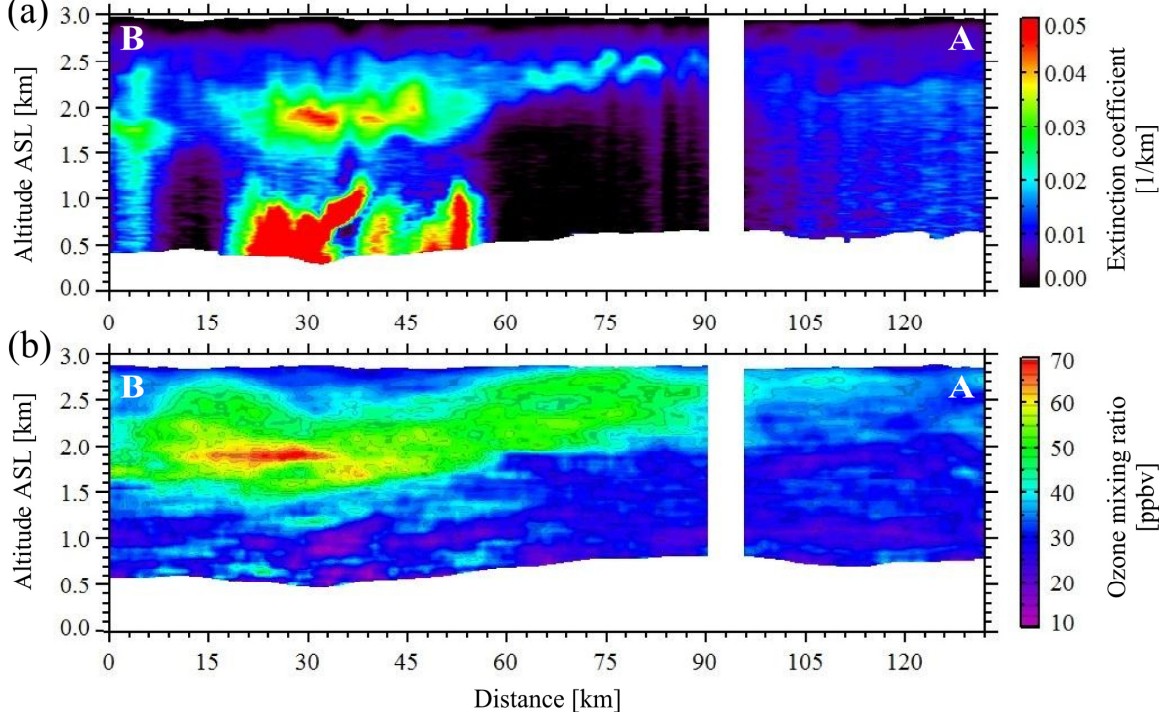

**Figure 12: (a) The aerosol extinction coefficient measured at a wavelength of 532 nm and (b) the O₃ mixing ratio derived from the lidar measurements for the flight segment A-B on August 24, 2013. The lidar measurements were collected between 2:48 and 3:18 PM local time (UTC – 6h). The height is above sea level (ASL). Distance is along the flight segment in Fig. 11. The blank section in this figure represents a region where clouds interfered with the lidar measurements.**



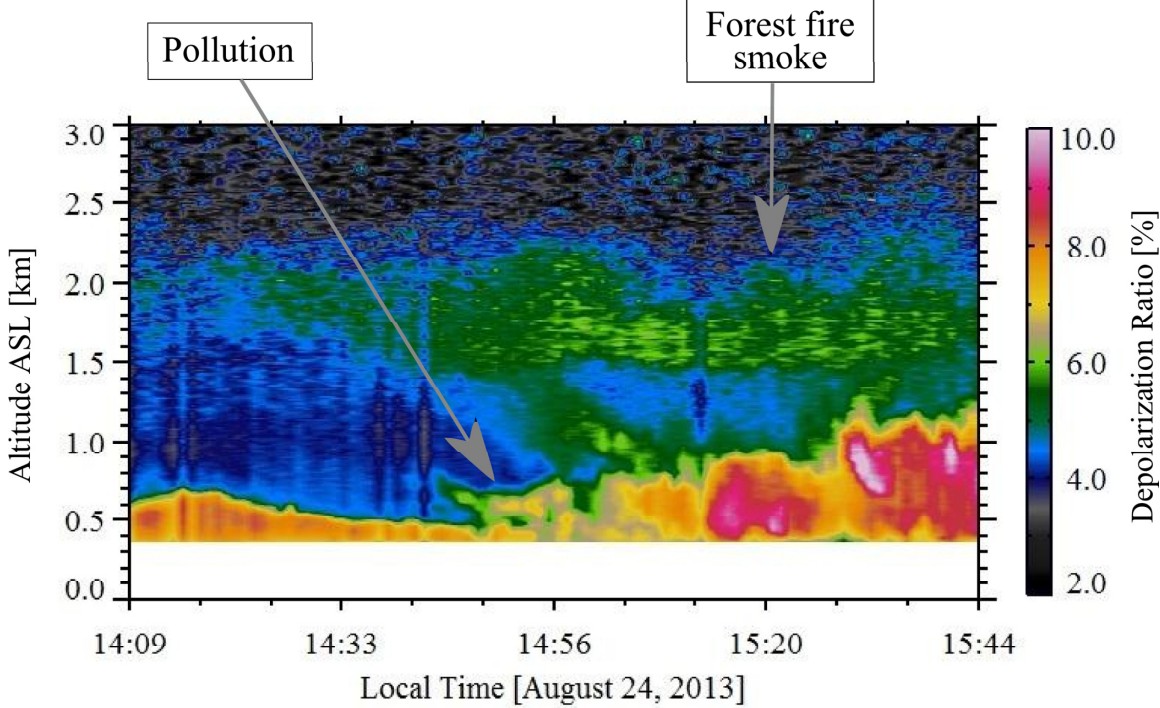

**Figure 13: Ground-based lidar depolarization ratio measurements acquired with the ECCC ground based lidar system located at 57.14° N, 111.6° W (AMS 13 in Fig. 11).**





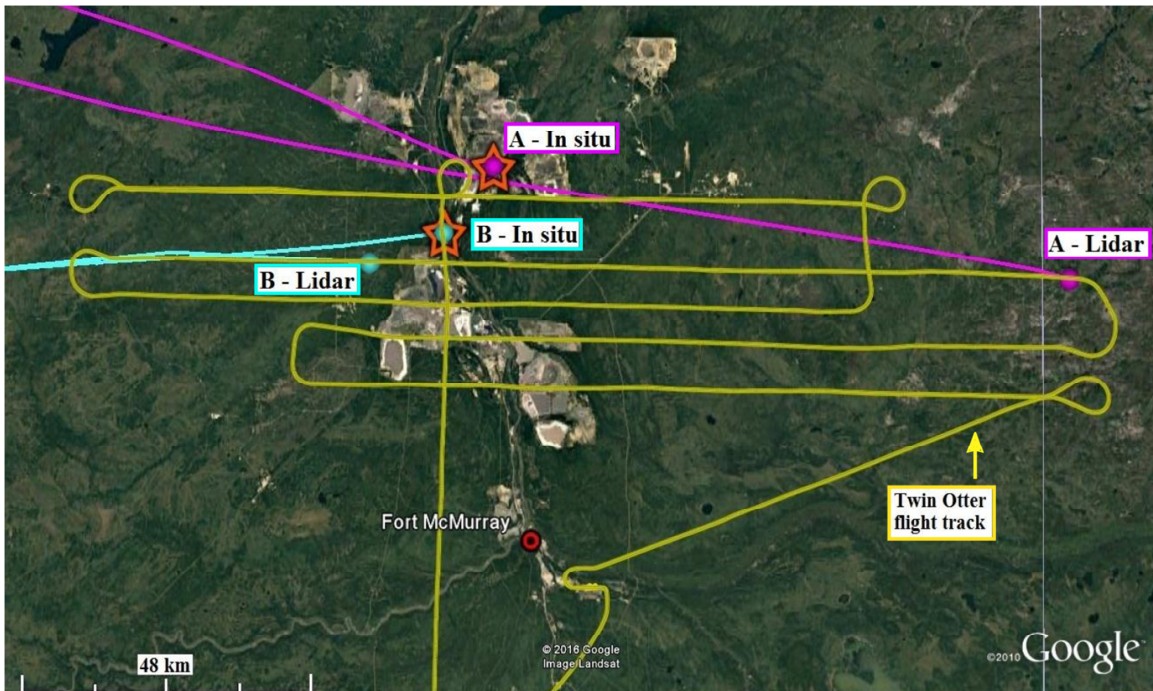

**Figure 14: Flight tracks of the Twin Otter aircraft (yellow lines) and locations the Convair spiral ascents for vertical profiles of in situ measurements (red stars). Backward trajectories coloured in blue and pink show the air coming from unpolluted and polluted areas respectively. Points labelled as A and B correspond to cases (a) and (b) in Fig. 15.**



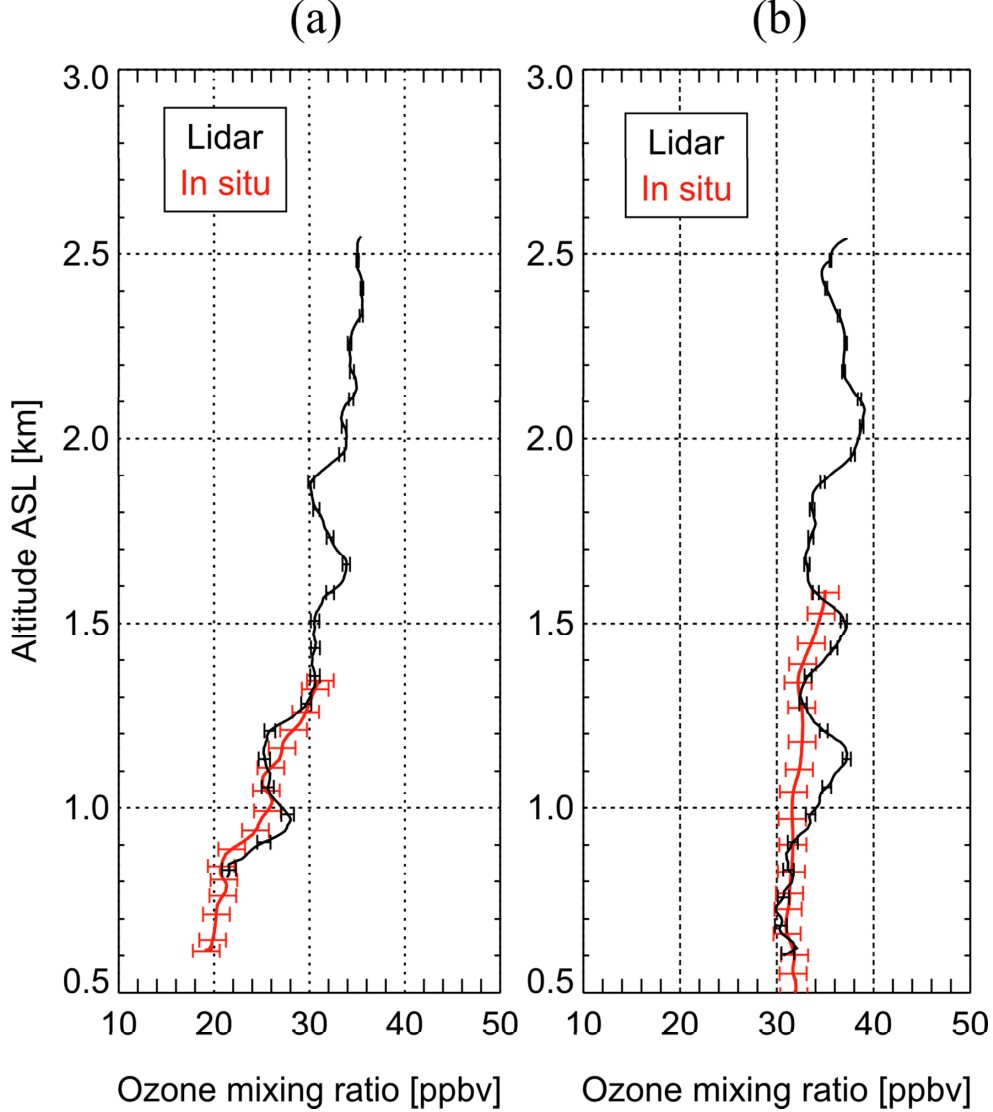

**Figure 15: A comparison between the in situ and the lidar derived ozone mixing ratio for the measurements taken in (a) polluted air and (b) unpolluted air. The locations of the measurements are indicated in Fig. 14 as points A and B for panels (a) and (b) respectively.**



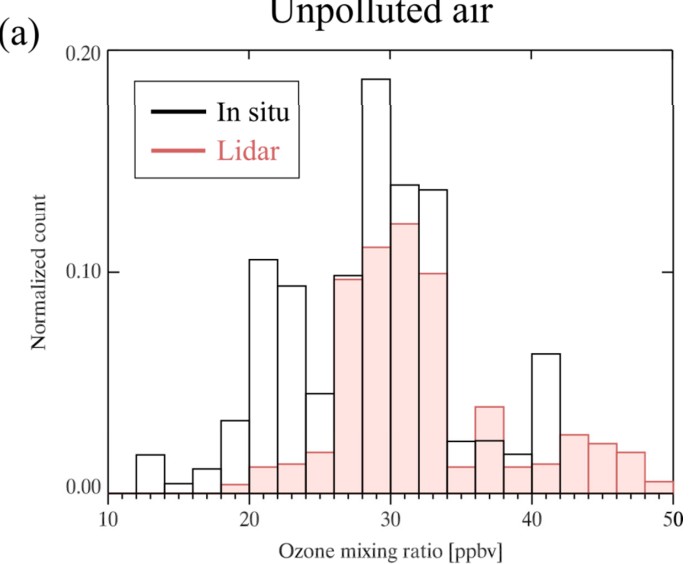

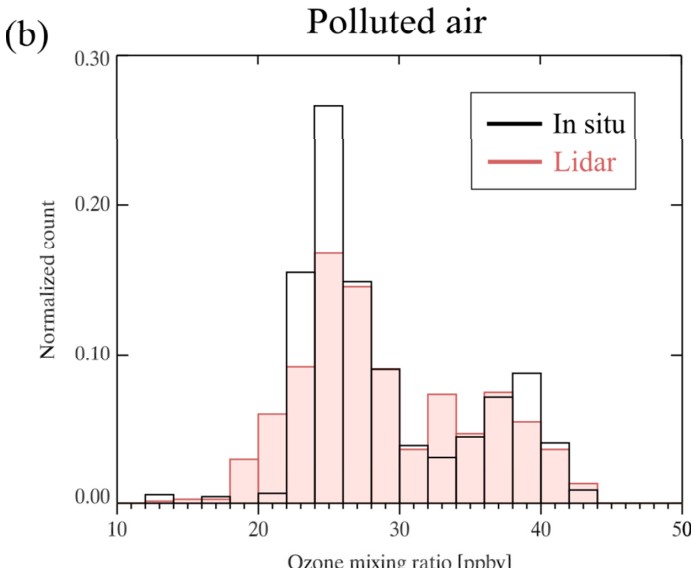

**Figure 16: A histogram of in situ and lidar derived ozone mixing ratio for the measurements taken within the boundary layer in (a) unpolluted air and (b) polluted air.**



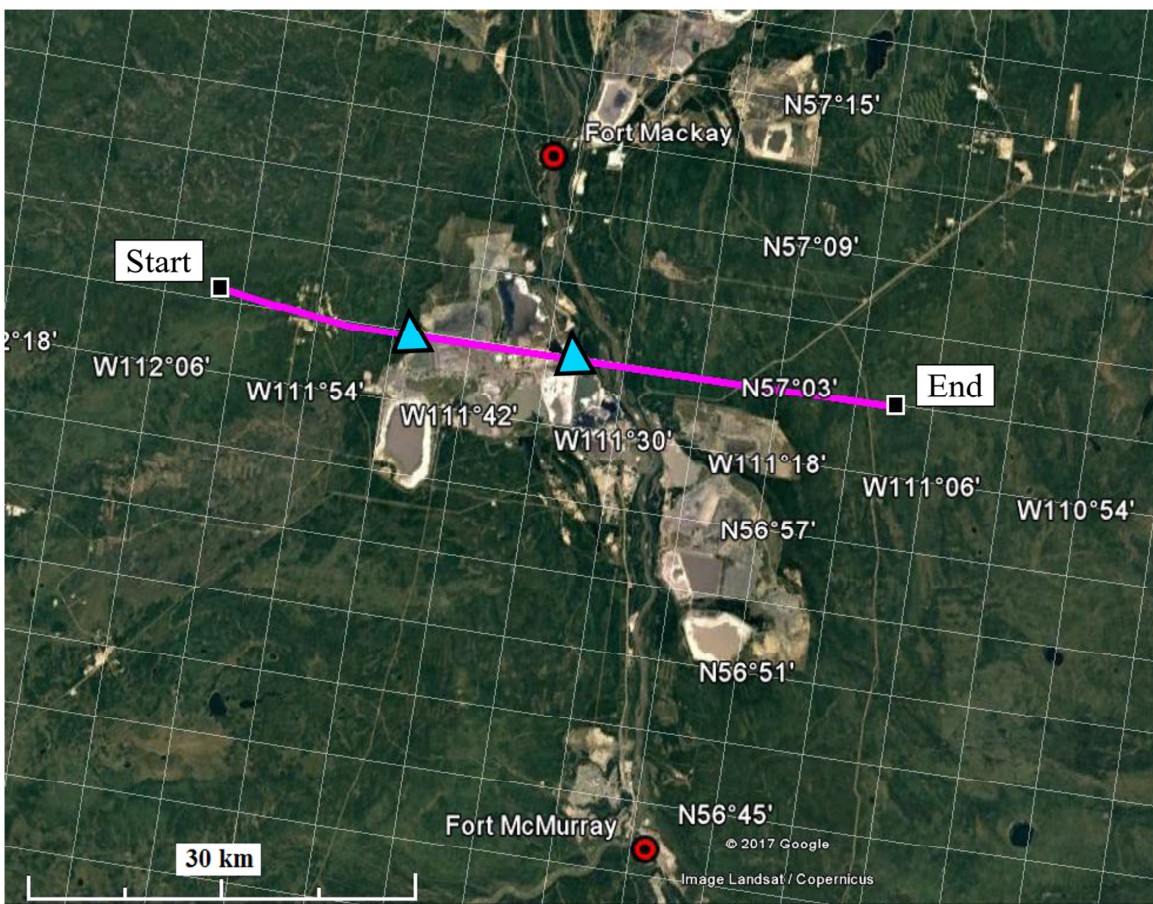

**Figure 17: A section of the Convair flight on August 23, 2013 is shown in pink. In situ measurements were collected from west to east between 11:27 and 11:37 AM local time (UTC – 6h) on August 23, 2013. The triangular marks indicate the positions of the dashed lines in Fig. 18.**



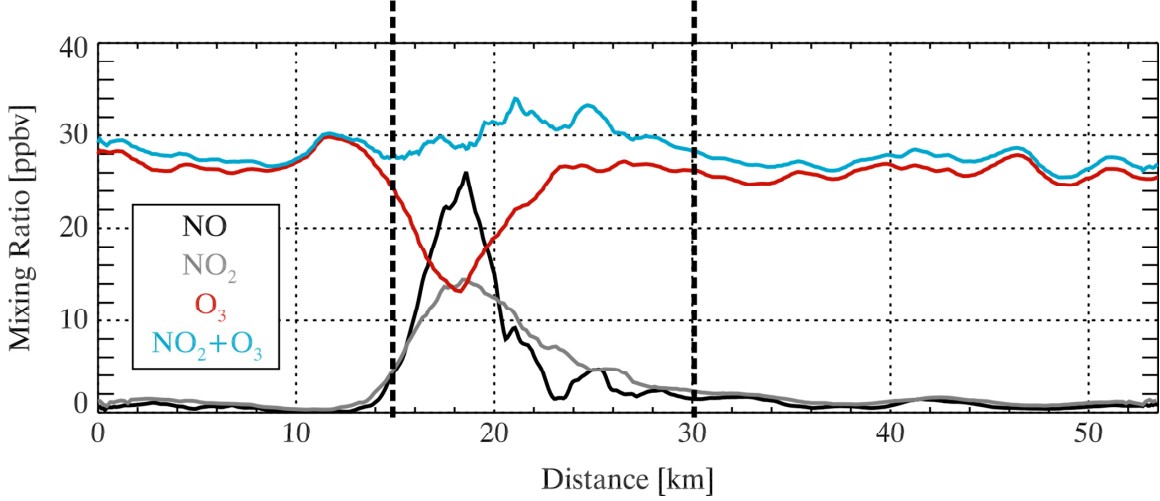

**Figure 18: In situ measurements of O₃, NO, and NO₂ taken from a section of the Convair flight on August 23, 2013 along the east-west direction over the oil sands industry and within the surface boundary layer. The in situ measurements were collected between 11:27 and 11:37 AM local time (UTC – 6h). The corresponding flight path is shown in Fig. 17. The dashed lines represent the positions of the blue triangles in Fig. 17.**



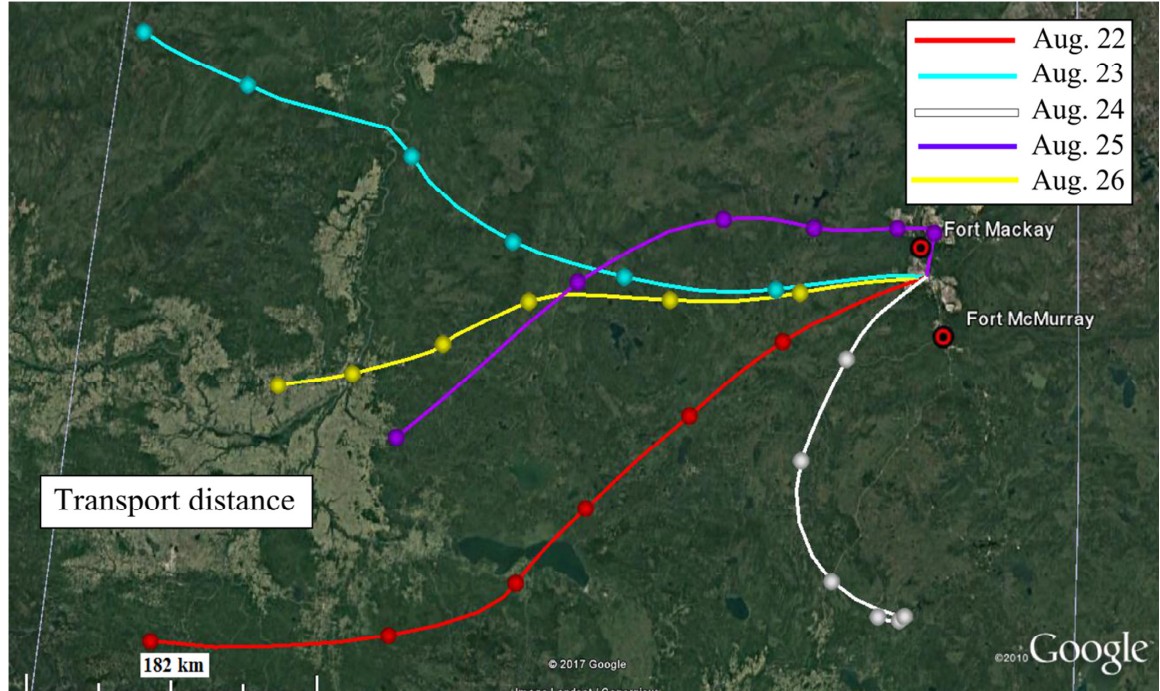

**Figure 19: Backward air trajectories initiated from an altitude of 700 m above sea level on August 22 to August 26, 2013 at 1 PM local time (UTC – 6h). The round mark along each trajectory**
5  **represents a time interval of 4 hours.**