# Peer review of "Airborne Lidar Measurements of Aerosol and Ozone Above the Canadian Oil Sands Region"

_Atmospheric Measurement Techniques, 2017_

## Referee Comment (RC1) · Anonymous Referee #1 · 6 Jan 2018

In their manuscript AMT-2017-391 titled "Airborne Lidar Measurements of Aerosol and Ozone Above the Canadian Oil Sands Region" Aggarwal et al. describe observations of ozone and aerosol concentrations made in summer 2013 with an airborne lidar over the Canadian oil sands region in northern Alberta. Their main finding is the lack of any ambient ozone enhancement downwind of the oil sands extraction and processing plants even though they represent a significant pollution source. This is an important finding that is contrary to many other studies that have found increased ozone concentrations downwind of urban areas or power plants. The only ozone enhancement the authors observed was in a forest fire plume that had been lofted above the boundary layer. In addition to the lidar observations, the authors use in situ measurements from another aircraft, observations from a ground-based aerosol and depolarization lidar, and HYSPLIT back trajectory analyses to support their conclusions. In the first part of the paper, the authors describe the lidar instrument and the ozone and aerosol retrieval technique, including an innovative approach to compute the differential aerosol backscatter and extinction correction terms making use of in situ aerosol size distribution and speciation measurements. Proper correction of the aerosol effects in the ozone lidar retrieval is critical because of the high aerosol concentrations that were observed above and downwind of the oil sands processing plants. The manuscript is written very clearly and the authors' reasoning is easy to follow. The conclusions presented in the manuscript are supported by the data and the figures and tables are all necessary, legible, and properly annotated. The topic of the paper fits well within the scope of AMT.

**I recommend publication after minor revisions.**

Specific comments:

page 5, lines 10 and 17:
UV signals were vertically smoothed over 45 m. The retrieved ozone profiles (Fig. 3b) appear to be reported at a much higher resolution. Error bars are given roughly every 100 m. Is that the effective resolution of the ozone profile observations (i.e. the separation of truly independent data points)? How did the authors compute the derivative of the logarithmic signal ratio? Least-square linear fit over multiple adjacent data points or Savitzky-Golay method? How were the two partial ozone profiles (AN 276/299 and PC 266/299) shown in Fig. 3b combined into one profile? Weighted averaging in the altitude region where both profiles overlap?

page 5, line 18:
What is the source of the air number density profile to convert ozone number density to mixing ratio? Ancillary pressure/temperature profile observations (e.g. nearby radiosondes)? Reanalysis data? Standard air number density profile?

page 6, line 2:
The authors need to describe briefly how their aerosol correction technique differs from other, previously published approaches. Also, include several references, e.g.:

Alvarez et al, 1998 (already in reference list)

Alvarez II, R. J., C. J. Senff, A. O. Langford, A. M. Weickmann, D. C. Law, J. L. Machol, D. A. Merritt, R. D. Marchbanks, S. P. Sandberg, W. A. Brewer, R. M. Hardesty, R. M. Banta, 2011: Development and application of a compact, tunable, solid-state airborne ozone lidar system for boundary layer profiling, *J. Atmos. Oceanic Technol.*, doi: 10.1175/JTECH-D-10-05044.1.

Browell, E. V., S. Ismail, and S. T. Shipley, 1985: Ultraviolet DIAL measurements of $O_3$ profiles in regions of spatially inhomogeneous aerosols, *Appl. Opt.*, **24**, 2827-2836.

Eisele and Trickl, 2005 (already in reference list)

page 6, line 20:
What reference height for aerosol extinction calibration do the authors typically use? Is it an altitude close to the aircraft (it appears that way from the aerosol extinction time/height plots, e.g. Fig 7a)?

page 7, line 1:
Only the aerosol particle refractive index is needed to compute $Q_{ext}$ and $Q_{back}$ from Mie theory. I don't understand why aerosol size distribution measurements are needed to calculate the efficiencies?

page 9, line 5:
"**As reported in the literature,** the aerosol correction is small (< 3 ppbv) …"

page 9, line 26:
"… **found in the literature** was 5 ppbv …

page 10, lines 4-15:
I double-checked the $SO_2$ interference estimates using the Brion et al. (1992-1998) $O_3$ and Vandaele Hermans, and Fally (2009) $SO_2$ absorption cross section data. I got interference terms of approximately 1, 5, and 35%*$SO_2$ concentration for the 266/299, 276/299, and 287/299 wavelength pairs, respectively. The 1% interference that I computed for the 266/299 pair would result in an ozone bias of up to 1.5 ppbv (larger than the 0.3 ppbv bias stated by the authors, but still quite small). Obviously, the magnitude of the interference terms depends on the choice of absorption cross section data. The $SO_2$ data in particular seem to vary quite a bit between the different published data sets. The authors need to state their sources of $O_3$ and $SO_2$ absorption cross section data and provide an error estimate of the interference terms due to absorption cross section data uncertainty.
The authors stated on page 5 that the 276/299 pair was used for $O_3$ profile retrieval to 1.8 km below the aircraft or about 1.1 km ASL (Fig. 3). On some flights, the boundary layer (with potential $SO_2$ concentrations of up to 30 − 150 ppbv) reached about 2 km ASL (Figs. 7a and 9a). Therefore, the 276/299 pair with its higher $SO_2$ sensitivity may have been used to retrieve the $O_3$ profile in the upper part of the boundary layer, which could have led to biases of up to several ppbv. Please clarify.

page 11, lines 11-12:

"… corrected by **fitting an** exponential decay function to …, where there **should** be no real optical signal."

page 12, line 25:
"… (distances from **40** to 60 km …"

page 13, lines 19-20:
"There is no evidence for increasing $O_3$ **for up to 10 hours downwind of the oil sands industrial areas.**"

page 15, line 1:
"**de**polarization ratio), …"

page 15, lines 11-12:
"… reveal a more **spherical shape** of forest fire particles."

page 15, lines 22-23:
"… and in situ measurements **with both aircraft sampling the same volume of air was not possible**."

page 17, line 10:
"… the dec**r**eased ozone abundance."

page 18, line 15:
Jacob et al., 2009 not in reference list

page 18, line 26:
"… result **of** meteorological conditions …"

page 18, lines 25-28:
Perhaps the absence of enhanced ozone downwind was at least in part due to low concentrations of suitable VOCs? Did the Convair aircraft measure VOCs?

References

Omit the following references since they are not mentioned in the text:
Angle et al, Calfapietra et al, Chu et al, Geddes et al, Haman et al., Ismail et al., Langford et al, 2011, Metha et al, Mie scattering source code, Permadi et al, Shephard et al, Wang et al, Yap et al

Page 25, line 21:
"…, 2010**b**."

Langford et al., 2010a, and Langford et al., 2010b are not in alphabetical order.

Figures

Fig. 4:
Change figure labeling: (b) Uncorrected, (c)-(e) Molecular (in black), Aerosol + Molecular (in red). Was the PC 266/299 pair not used in this case?

Fig. 5: Indicate location of main emission sources.

Fig. 12a: Near 15 km and between 60 and 90 km distance, the extinction coefficient is near 0 within the boundary layer. What caused this? Aerosol fluctuations in the reference region?

---

## Referee Comment (RC2) · Anonymous Referee #2 · 22 Mar 2018

Aggarwall et al. (AMT-2017-391) mainly employs an airborne aerosol and ozone lidar to measure air pollution from the Canadian oil sands extraction industry in northern Alberta, as well as to measure a separate fire-smoke layer over the area. The major lidar technique included in this paper is well defined and explained. The interpretation of the observation phenomenon within the industrial-polluted boundary layer as NOx titration is adequately supported for a measurement paper and the ozone production in the layer aloft is reasonable enough. The technical aspect of properly accounting for the aerosol influence on the ozone retrieval is very useful and should be helpful for future lidar measurements in smoke plumes. The conclusions are clear and succinct. I recommend publication following revisions. General considerations The authors should

discuss the actinic flux field in conjunction with the discussions of transmittance, absorption, scattering, photochemical formation, etc. What was the cloud condition? Was it cloudy, hazy, humid? All of these MET variables would make a difference when considering ozone production/destruction. Did you consider potential NOx contributions from the aircraft affecting the titration? Perhaps the altitude was always above the PBL? My most significant concern is that this article lacks a discussion of the uncertainty budget calculation from different sources (statistical, background correction, aerosol correction, Rayleigh correction, differential ozone absorption cross section, e.g., Wang et al., 2017) for the ozone DIAL measurement. This discussion is essential. Can you quantify or at least estimate the vertical (spatial) resolution for the ozone lidar, which is an important parameter for profiling instruments. The vertical resolution is closely related to your signal/data processing. Specific considerations P1, L26: "This paper concerns the methodology and results of airborne lidar measurements of aerosol and ozone...". However, the Introduction section does not provide any review of the instrumentation or retrieval technique of either airborne aerosol or ozone lidars. P3, L8: What are the conversion efficiencies and final pulse energy for the three wavelengths?

P5, L19-21: The authors choose to retrieve ozone separately from analog and PC channels while a more common approach is to merge the analog and PC signals first at a reference counting rate (e.g. Kuang et al., 2011). So, do you then merge the ozone profiles? Can you explain more about how and where you exactly merge the ozone profile, in a constant altitude range or at a single point? P 6.L10-11: What is the potential error in retrieved ozone amounts from assuming consistent aerosol composition and size distribution throughout the boundary layer? P7 L 8-11: The author use the refractive index of kaolinite to compute the extinction and backscatter coefficients based on the studies, that kaolinite to be the prominent clay particle in the oil sands region. Here the quantitative value (e.g. fraction) for the "prominent" role would be better if provided. Do you have any evidence that the aerosols are actually Kaolinite? What is the potential error in retrieved ozone amounts if their composition (and complex refractive index) are different? P7, L21: "not particle size", the lidar ratio

also varies with aerosol type, or refractive index, and probably humidity, not only size distribution. P11 L21: The GDAS meteorological dataset for HYSPLIT input has two resolution options: 1degree and 0.5degree. Which option did the authors select? For an aircraft measurement up to 150km downwind of emission, would the resolution influence the trajectory accuracy? Did the author consider other options such as NAM (12km) and HRRR (3km), perhaps? P 19.L 1-8: This paragraph lacks scientific analysis. The lack of supporting data such ozone precursors measurement in the fire plume to indicate the chemical production mechanism of fire ozone formation. The authors quote the paper by Jaffe and Wigder, 2012, but didn't provide any further analysis about the influence factors for ozone production (fire emissions, efficiency of combustion, photochemical reactions, aerosol effects on chemistry and radiation), meteorological patterns were mentioned without quantitative analysis. The meaning of "the temperature would have been greater in the plumes above the fires" is not clear. This paragraph requires quantitative estimation of the environmental variables. Minor issues P. 1, L 17: "ground-based lidar" P. 3, L 16: Identify PMT model(s) Pg. 3, L 21: Why isn't the 532 smoothing distance an integer multiple of the range bin 23=6.13*3.75? P 6.L 15: "523nm" should be "532nm". P6, equation (3) and (4): did you say anywhere in the context "m" represent the refractive index? P8, L19-20: how did you calculate the extinction and backscatter profiles at "the UV wavelengths" based on the profiles at green? Did you assume a value for Angstrom exponent? P10, L6-12: SO2 absorption cross section may significantly vary with database. Can you give the reference for the source of the SO2 absorption cross section. P11, L15: there should be a newer reference than (Draxler and Hess, 1998) for HYSPLIT. Kuang, S., J. F. Burris, M. J. Newchurch, S. Johnson, and S. Long (2011), Differential Absorption Lidar to measure subhourly variation of tropospheric ozone profiles, IEEE Trans. Geosci. Remote Sens., 49, 557-571, doi: 10.1109/TGRS.2010.2054834. Wang, L., Newchurch, M. J., Alvarez II, R. J., Berkoff, T. A., Brown, S. S., Carrion, W., DeYoung R. J., Johnson, B. J., Ganoe, R., Gronoff, G., Kirgis G., Kuang, S., Langford, A. O., Leblanc T., McDuffie E. E., McGee, T. J., Pliutau, D., Senff, C. J., Sullivan, J. T.,

Sumnicht, G., Twigg, L. W., & Weinheimer, A. J. (2017). Quantifying TOLNet ozone lidar accuracy during the 2014 DISCOVER-AQ and FRAPPE campaigns. Atmospheric Measurement Techniques, 10(10), 3865-3876.

Please also note the supplement to this comment:
https://www.atmos-meas-tech-discuss.net/amt-2017-391/amt-2017-391-RC2-supplement.pdf
* * *

---

## Author Comment (AC1) · 18 May 2018

**Manuscript 2017-391: Aggarwal et al., Airborne Lidar Measurements …**

**Response to Comments from Reviewer 1**

Note: changes in the revised manuscript have been highlighted in yellow

**General Comment from the Authors**

The authors appreciate the meticulous work by both reviewers. Substantial improvements have been made in response.

The first author recently completed her Ph.D. examination and through that process there have been a few improvements to the analysis that were not requested by the reviewers.

a) Equation 5 of the submitted manuscript included an approximation in terms of the effective radius. This was not necessary. This equation now has the integral over the size spectrum as for the other calculations in equations (3), (4), (6) and (7). The explanation is now easier to follow and the calculations are more accurate. The calculated aerosol corrections are slightly different, but the results have not changed.

b) The temporal averaging was previously 1.3 minutes for some figures and 3 minutes for others. The temporal averaging is now consistent at 1.5 minutes for all of the $O_3$ analysis in the figures. This corresponds to a distance of about 7.5 km along the flight track. This had an impact on the histograms of Fig. 16 since previously the averaging was 3 minutes. Now the averaging is 1.5 minutes for both the lidar and in situ measurements represented in Fig. 16. The description of the histograms has been changed slightly on page 18, lines 7 to 25.

*Reviewer 1, Comment 1*

*UV signals were vertically smoothed over 45 m. The retrieved ozone profiles (Fig. 3b) appear to be reported at a much higher resolution. Error bars are given roughly every 100 m. Is that the effective resolution of the ozone profile observations (i.e. the separation of truly independent data points)? How did the authors compute the derivative of the logarithmic signal ratio? Least-square linear fit over multiple adjacent data points or Savitzky-Golay method? How were the two partial ozone profiles (AN 276/299 and PC 266/299) shown in Fig. 3b combined into one profile? Weighted averaging in the altitude region where both profiles overlap?*

**Response to R1, Comment 1**

An improved description is now provided.  See page 5, line 27 to page 6 line 15.

*R1, Comment 2*

*What is the source of the air number density profile to convert ozone number density to mixing ratio? Ancillary pressure/temperature profile observations (e.g. nearby radiosondes)? Reanalysis data? Standard air number density profile?*

**Response to R1, Comment 2**

Radiosondes launched from Edmonton. Now mentioned on page 6, line 9.

**R1, Comment 3**

*The authors need to describe briefly how their aerosol correction technique differs from other, previously published approaches. Also, include several references.*

**Response to R1, Comment 3**

Description of the difference has been added on page 6, lines 20 to 28.

**R1, Comment 4**

*What reference height for aerosol extinction calibration do the authors typically use? Is it an altitude close to the aircraft (it appears that way from the aerosol extinction time/height plots, e.g. Fig 7a)?*

**Response to R1, Comment 4**

The reference height was a minimum of 200 m above the top of the boundary layer. This has been added to the manuscript on page 7, line 17.

**R1, Comment 5**

*Only the aerosol particle refractive index is needed to compute Qext and Qback from Mie theory. I don't understand why aerosol size distribution measurements are needed to calculate the efficiencies. (?)*

**Response to R1, Comment 5**

$Q_{ext}$ and $Q_{back}$ depend on particle size. The following figure shows a calculation of $Q_{ext}$ based on an aerosol size distribution used in this study.

[Figure]

***R1, Comment 6***

*I double-checked the SO2 interference estimates using the Brion et al. (1992-1998) O3 and Vandaele Hermans, and Fally (2009) SO2 absorption cross section data. I got interference terms of approximately 1, 5, and 35%\*SO2 concentration for the 266/299, 276/299, and 287/299 wavelength pairs, respectively. The 1% interference that I computed for the 266/299 pair would result in an ozone bias of up to 1.5 ppbv (larger than the 0.3 ppbv bias stated by the authors, but still quite small). Obviously, the magnitude of the interference terms depends on the choice of absorption cross section data. The SO2 data in particular seem to vary quite a bit between the different published data sets. The authors need to state their sources of O3 and SO2 absorption cross section data and provide an error estimate of the interference terms due to absorption cross section data uncertainty.*

**Response to R1, Comment 6**

The text has been changed in the paragraph starting on page 11, line 17 in response to the referee's comment. We now provide a range of values for bias due to $SO_2$ absorption based on three separate sources of absorption cross sections.

***R1, Comment 7***

*The authors stated on page 5 that the 276/299 pair was used for O3 profile retrieval to 1.8 km below the aircraft or about 1.1 km ASL (Fig. 3). On some flights, the boundary layer (with potential SO2 concentrations of up to 30 – 150 ppbv) reached about 2 km ASL (Figs. 7a and 9a). Therefore, the 276/299 pair with its higher SO2 sensitivity may have been used to retrieve the O3 profile in the upper part*

**Response to R1, Comment 7**

The transition was at a height of at least 300 m above the boundary layer. A better description is now provided on page 6 lines 9 to 15.

***R1, Comment 8***

*Perhaps the absence of enhanced ozone downwind was at least in part due to low concentrations of suitable VOCs? Did the Convair aircraft measure VOCs?*

**Response to R1, Comment 8**

The measurements of VOCs on the Convair aircraft are beyond the scope of this study. Those measurements will be taken into account in a different study by a separate group of researchers (not published in this issue) in which the Canadian Air Quality forecast model will be used to assess the production/destruction of ozone.

*R1, Comment 9*
*Fig. 4*
*Change figure labeling: (b) Uncorrected, (c)-(e) Molecular (in black), Aerosol + Molecular (in red). Was the PC 266/299 pair not used in this case?*

**Response to R1, Comment 9**
See the new caption for Fig. 4.

*R1, Comment 10*
*Fig. 5: Indicate location of main emission sources.*

**Response to R1, Comment 10**
A box has been drawn around the region of oil sands industrial emissions.

*R1, Comment 11*
*Fig. 12a: Near 15 km and between 60 and 90 km distance, the extinction coefficient is near 0 within the boundary layer. What caused this? Aerosol fluctuations in the reference region*

**Response to R1, Comment 11**
Yes, there was an issue with aerosol fluctuations in the reference region and this has been fixed.

---

## Author Comment (AC2) · 18 May 2018

**Manuscript 2017-391: Aggarwal et al., Airborne Lidar Measurements …**

**Response to Comments from Reviewer 2**

Note: changes in the revised manuscript have been highlighted in yellow

**General Comment from the Authors**

The authors appreciate the meticulous work by both reviewers. Substantial improvements have been made in response.

The first author recently completed her Ph.D. examination and through that process there have been a few improvements to the analysis that were not requested by the reviewers.

a) Equation 5 of the submitted manuscript included an approximation in terms of the effective radius. This was not necessary. This equation now has the integral over the size spectrum as for the other calculations in equations (3), (4), (6) and (7). The explanation is now easier to follow and the calculations are more accurate. The calculated aerosol corrections are slightly different, but the results have not changed.

b) The temporal averaging was previously 1.3 minutes for some figures and 3 minutes for others. The temporal averaging is now consistent as 1.5 minutes for all of the analysis in figures. This corresponds to a distance of about 7.5 km along the flight track. This had an impact on the histograms of Fig. 16 since previously the averaging was 3 minutes. Now the averaging is 1.5 minutes for both the lidar and in situ measurements represented in Fig. 16. The description of the histograms has been changed slightly on page 18, lines 7 to 25.

*Reviewer 2, Comment 1*

*The authors should discuss the actinic flux field in conjunction with the discussions of transmittance, absorption, scattering, photochemical formation, etc. What was the cloud condition? Was it cloudy, hazy, humid? All of these MET variables would make a difference when considering ozone production/destruction.*

**Response to R1, Comment 1**

Radiative flux was not measured as part of this study. There was some consideration of the conditions in the discussion section from page 19, line 25 to page 20, line 16. We have also added a brief reference to the weather conditions on page 13, line 22.

There is a separate study in progress by a separate group of researchers in which the Canadian Air Quality forecast model is being used to assess the production/destruction of ozone. The will properly take into account all of the factors for ozone production and destruction. The results of the current study will inform the model development.

***R2, Comment 2***

*Did you consider potential NOx contributions from the aircraft affecting the titration? Perhaps the altitude was always above the PBL?*

**Response to R2, Comment 2**

The Twin Otter was flying above the boundary layer except for take off and landing. It was very unlikely that the exhaust from the Convair would have been intercepted by the lidar measurements and would have been a very small contribution in comparison to the industrial emissions.

**R2, Comment 3**

*My most significant concern is that this article lacks a discussion of the uncertainty budget calculation from different sources (statistical, background correction, aerosol correction, Rayleigh correction, differential ozone absorption cross section, e.g., Wang et al., 2017) for the ozone DIAL measurement. This discussion is essential.*

**Response to R2, Comment 3**

The largest contribution to the uncertainty in the derived ozone mixing ratio is from the aerosol correction. One reason is that the size distribution of the aerosol particles is not constant with height or horizontal distance. The particle effective radius (area averaged) was found to vary between 0.06 and 0.08 micrometers along a flight through the polluted air above the oil sands. Another source of uncertainty was that the aerosol composition, and thus refractive index, was not known. We had assumed kaolinite in the submitted manuscript since most of the aerosol appeared to be the mineral dust that was emitted due to the surface mining activities.

In response to the reviewers' comment we have computed the aerosol correction for a range of measured particle size distributions and for a variety of aerosol compositions including kaolinite, diesel soot, sulfuric acid, toluene SOA, and ammonium sulfate. Figure 4 now shows a range of aerosol corrections corresponding to the various compositions and particle size distributions. The vertical profiles of derived ozone are now shown as a range that represents the uncertainty due to the aerosol correction. Description of this is found in the manuscript from page 9, line 18 to page 10, line 19

We also now give more attention to the uncertainty due to the bias due to interference by differential absorption of $SO_2$. The text has been changed in the paragraph starting on page 11, line 17 in response to the referee's comment. We provide a range of values for bias due to $SO_2$ absorption based on three separate sources of absorption cross sections.

The amount of ozone measured by the lidar is within the range of in situ measurements on the Convair aircraft and at ground sites. There is no discrepancy to explain.

**R2, Comment 4**

*Can you quantify or at least estimate the vertical (spatial) resolution for the ozone lidar, which is an important parameter for profiling instruments. The vertical resolution is closely related to your signal/data processing.*

**Response to R2, Comment 4**

This is now stated more clearly in the paragraph starting on page 5, line 21.

*R2, Comment 5*

*"This paper concerns the methodology and results of airborne lidar measurements of aerosol and ozone …". However, the introduction section does not provide any review of the instrumentation or retrieval technique of either airborne aerosol or ozone lidars.*

**Response to R2, Comment 5**

As this paper was not intended to provide a review, the word "methodology" has been changed to "measurement technique" on page 1, line 26.

Publications concerning previous lidar studies were cited throughout the manuscript where it was appropriate.

*R2, Comment 6*

*What are the conversion efficiencies and final pulse energy for the three wavelengths?*

**Response to R2, Comment 6**

This is now included in the manuscript on page 3, lines 8 to 13.

*R2, Comment 7*

*The authors choose to retrieve ozone separately from analog and PC channels while a more common approach is to merge the analog and PC signals first at a reference counting rate (e.g. Kuang et al., 2011). So, do you then merge the ozone profiles? Can you explain more about how and where you exactly merge the ozone profile, in a constant altitude range or at a single point?*

**Response to R2, Comment 7**

A better description is now provided on page 6, lines 9 to 15.

*R2, Comment 8*

*What is the potential error in retrieved ozone amounts from assuming consistent aerosol composition and size distribution throughout the boundary layer?*

*The author use the refractive index of kaolinite to compute the extinction and backscatter coefficients based on the studies, that kaolinite to be the prominent clay particle in the oil sands region. Here the quantitative value (e.g. fraction) for the "prominent" role would be better if provided. Do you have any evidence that the aerosols are actually Kaolinite? What is the potential error in retrieved ozone amounts if their composition (and complex refractive index) are different?*

**Response to R2, Comment 8**

We have made some changes in response to this referee comment. The variations in particle size distribution and assumption of aerosol composition are actually the main source of uncertainty in measured $O_3$ concentration when there is substantial aerosol. In the revised manuscript we are providing an estimate of this uncertainty by showing a range in the magnitude of correction for a variety of aerosol compositions and particle size distributions.

See the description from page 9, line 18 to page 10 line 19

*R2, Comment 9*

*"not particle size", the lidar ratio also varies with aerosol type or refractive index and probably humidity, not only size distribution.*

**Response to R2, Comment 9**

The lidar ratio was taken into account when assessing different aerosol types as in the response to the previous comment.

*R2, Comment 10*

*The GDAS meteorological dataset for HYSPLIT input has two resolution options: 1degree and 0.5degree. Which option did the authors select? For an aircraft measurement up to 150km downwind of emission, would the resolution influence the trajectory accuracy?*

**Response to R2, Comment 10**

We used the 1 degree resolution. Now mentioned on page 13, line 10.

***R2, Comment 11***

*This paragraph lacks scientific analysis. The lack of supporting data such ozone precursors measurement in the fire plume to indicate the chemical production mechanism of fire ozone formation. The authors quote the paper by Jaffe and Wigder, 2012, but didn't provide any further analysis about the influence factors for ozone production (fire emissions, efficiency of combustion, photochemical reactions, aerosol effects on chemistry and radiation), meteorological patterns were mentioned without quantitative analysis.*

*The meaning of "the temperature would have been greater in the plumes above the fires" is not clear. This paragraph requires quantitative estimation of the environmental variables.*

**Response to R2, Comment 11**

Measurements were not obtained at the fires. All we have are the measurements of enhanced ozone in the forest fire smoke. There is no data beyond that to form the basis for a more in-depth analysis. The point of including the observation was that it stands in contrast to the lack of ozone production in the industrial pollution. The reference to temperature is actually indirectly implying that there would be greater concentration of VOCs in the fire emissions. As we don't have measurements of VOC concentrations in the fire emissions, and as the reviewer objects to the speculation, we have removed the reference to temperature associated with forest fires.

***R2, Comment 12***

*Identify PMT model(s)*

**Response to R2, Comment 12**

Added to page 3, line 21

***R2, Comment 13***

*Why isn't the 532 smoothing distance an integer multiple of the range bin 23=6.13\*3.75?*

**Response to R2, Comment 13**

That was an error. See the paragraph starting on page 5, line 21.

***R2, Comment 14***

*Equation (3) and (4): did you say anywhere in the context "m" represents the refractive index?*

**Response to R2, Comment 14**

Added on page 8, line 2.

**R2, Comment 15**

*How did you calculate the extinction and backscatter profiles at "the UV wavelengths" based on the profiles at green? Did you assume a value for Angstrom exponent?*

**Response to R2, Comment 15**

Please see all of Section 3.1. We are using the 532 nm lidar measurements and the in situ measurements of particle size distribution to determine the extinction and backscatter coefficient profiles at the UV wavelengths. This analysis does not require an Angstrom exponent.

**R2, Comment 16**

*SO2 absorption cross section may significantly vary with database. Can you give the reference for the source of the SO2 absorption cross section.*

**Response to R2, Comment 16**

More discussion of this is now given in the paragraph starting on page 11, line 17.

**R2, Comment 17**

*There should be a newer reference than (Draxler and Hess, 1998) for HYSPLIT*

**Response to R2, Comment 17**

This was replaced with a more recent reference. Page 13, line 4.

Stein, A. F., Draxler, R. R., Rolph, G. D., Stunder, B. J. B., Cohen, M. D., and Ngan, F.: NOAA's HYSPLIT atmospheric transport and dispersion modeling system, B. Am. Meteorol. Soc., **96**, 2059-2077, doi:10.1175/BAMS-D-14-00110.1, 2015.